

# Development, Characterization and Rapid Diagnostics of an Aircraft Aerosol Mass Spectrometer Inlet System

Dongwook Kim[1], Pedro Campuzano-Jost[1], Hongyu Guo[1], Douglas A. Day[1], Da Yang[1,2], Suresh
Dhaniyala[2], Leah Williams[3], Philip Croteau[3], John Jayne[3], Douglas Worsnop[3,4], Rainer Volkamer[1] and
Jose L. Jimenez[1]

[1]Department of Chemistry and CIRES, University of Colorado Boulder, Boulder, CO, 80309-0215, USA
[2]Department of Mechanical Engineering, Clarkson University, Potsdam, NY, 13699, USA
[3]Aerodyne Research Inc., Billerica, MA, 01821, USA
[4]Institute for Atmospheric and Earth System Research/Physics, Faculty of Science, University of Helsinki, Helsinki, 00014,
Finland

*Correspondence to:* Jose L. Jimenez (jose.jimenez@colorado.edu)

**Abstract.** Field-deployable real-time aerosol mass spectrometers typically use an aerodynamic lens as an inlet that collimates aerosols into a narrow beam over a wide range of particle sizes. Such lenses need constant upstream pressure to work consistently. Deployments in environments where the ambient pressure changes, *e.g.,* on aircraft, typically use pressure-controlled inlets (PCI). These have performed less well for supermicron aerosols, such as the larger particles in stratospheric
air and some urban hazes. In this study, we developed and characterized a new PCI design ("CU PCI-D") coupled with a recently developed $PM_{2.5}$ aerodynamic lens, with the goal of sampling the full accumulation mode of ambient aerosols with minimal losses up to upper troposphere and lower stratosphere (UTLS) altitudes. A new computer-controlled lens alignment system and a new 2D particle beam imaging device that improves upon the Aerodyne aerosol beam width probe (BWP) have been developed and tested. These techniques allow for fast automated aerosol beam width and position measurements and
ensure the aerodynamic lens is properly aligned and characterized for accurate quantification, in particular for small sizes that are hard to access with monodisperse measurements. The CU PCI-D was tested on the TI³GER campaign aboard the NCAR/NSF G-V aircraft. Based on comparisons with the co-sampling UHSAS particle sizer, the CU aircraft AMS with the modified PCI consistently measured ~89% of the accumulation mode particle mass in the UTLS.





## 1. Introduction

Aerosols play an important role in the atmosphere's radiative balance via direct and indirect forcing (IPCC, 2014; Seinfeld and Pandis, 2016). In the upper troposphere, condensable vapors formed from convected precursors create new particles, helped by the low temperature and low condensational sink, providing cloud condensation nuclei to the lower troposphere (Williamson et al., 2019). Stratospheric aerosols have a significant radiative forcing due to their long lifetime. Most stratospheric aerosols are sulfate or carbonaceous. A minor portion of stratospheric aerosols originate from meteors and spacecraft (Murphy et al., 1998, 2023). Some geoengineering proposals suggest the injection of sulfur into the stratosphere to form sulfate aerosols that would scatter incoming solar radiation and combat global warming (Crutzen, 2006; Robock et al., 2009; Keith et al., 2016). Aerosols also provide a medium for heterogeneous reactions that enhance the reactive chlorine budget, which causes ozone destruction (Fahey et al., 1993). More recently, Solomon et al. (2023) suggested that organic aerosols originating from biomass burning enhance stratospheric chlorine activation, leading to further ozone destruction. Annual mean ozone in the lower stratosphere (LS; from tropopause to ~30 hPa) has decreased for the last few decades, a trend not captured by models (Ball et al., 2020). Recent studies have suggested that short-lived halogen compounds (Villamayor et al., 2023), including iodine cycling between the both gas and particle phases (Koenig et al., 2020), may contribute to ozone destruction in the LS. However, quantitative observations of the chemical composition of stratospheric aerosol by in-situ instruments are rare due to the challenges of reaching and operating at these high altitudes.

Field-deployable aerosol mass spectrometers that measure aerosol chemical composition in real-time using a specialized inlet, typically an aerodynamic lens (ADL), which focuses particles over a relatively large range of sizes into a narrow particle beam with minimal losses. These spectrometers include the Aerodyne Aerosol Mass Spectrometer (AMS, Aerodyne Research) (Canagaratna et al., 2007), single particle mass spectrometers such as Particle Analysis by Laser Mass Spectrometry (PALMS), Aerosol Time-of-flight Mass Spectrometry (ATOFMS), and Aircraft-based Laser ABlation Aerosol MAss spectrometer (ALABAMA) (Pratt et al., 2009; Clemen et al., 2020; Jacquot et al., 2024), and the chemical analysis of aerosol online (CHARON) instrument (Müller et al., 2017; Piel et al., 2019) The standard Liu type lens (also commonly referred to as the PM$_1$ lens where PM$_1$ means particulate matter below 1 µm diameter) has been the most widely used ADL in the past decades (Liu et al., 1995a, 1995b, 2007; Zhang et al., 2002, 2004b). The particle transmission efficiency (TE) through an ADL depends not only on the physical design, but also on the operating pressures, the size of the critical orifice used upstream of the ADL, and the solid angle of the particle beam that overlaps with the detector area (Huffman et al., 2005; Murphy, 2007). A particle relaxation chamber can be added to reduce particle losses after critical orifices (Wang and McMurry 2007). The PM$_1$ lens transmits most particles of ~50–800 nm vacuum aerodynamic diameter ($d_{va}$) range, encompassing most of the accumulation mode aerosols in the troposphere (Guo et al., 2021). However, the PM$_1$ lens can miss a significant fraction of aerosol mass when the accumulation mode grows very large *e.g.,* in highly polluted environments (Elser et al., 2016) and in the stratosphere (Brock et al., 2019; Guo et al., 2021). Two other ADL designs developed at Aerodyne, the high-pressure lens (HPL) (Williams et al., 2013) and the PM$_{2.5}$ lens (Peck et al., 2016; Xu et al., 2017), significantly extended the





transmittable aerosol size range beyond 1 μm $d_{va}$ by increasing the operating pressures. A custom-designed ADL with conical-shaped orifices enabled supermicron aerosol sampling in the ALABAMA instrument (Clemen et al., 2020).

In an AMS, the particle beam collimated by an ADL is flash-vaporized on a porous tungsten inverted cone (standard vaporizer, SV; 3.8 mm OD) at ~600°C. The depth of the inverted cone is 4 mm (Hu et al., 2017a). The vaporized molecules can be ionized by electron ionization (EI, ~70 eV) and detected by time-of-flight mass spectrometry (Canagaratna et al., 2007). Both the thermal vaporization and the hard ionization often cause molecular fragmentation. Analysis of the fragmentation patterns allows characterization of organic aerosols (*e.g.,* oxygenated versus hydrocarbon-like organic aerosol) (Ng et al.,

2011) and in some cases sulfate (*e.g.,* inorganic versus organic sulfate) (Chen et al., 2019; Song et al., 2019; Schueneman et al., 2021). Apportionment of organic vs. inorganic nitrate has been attempted based on the low observed $NO_x^+$ ratio ($NO_2^+/NO^+$) from organic nitrates compared to $NH_4NO_3$ (Farmer et al., 2010; Day et al., 2022). Day et al. (2022) summarized the AMS instrumental variabilities of the $NO_x^+$ ratio from $NH_4NO_3$ and organic nitrates and demonstrated that the $NO_x^+$ ratio variability from organic nitrates can be corrected by the $NO_x^+$ ratio of $NH_4NO_3$. While the particles (or vaporized gases) likely

interact more with the hot vaporizer surface when the particle beam impacts near the center of SV due to its conical geometry, this effect has not been thoroughly investigated. It is typically assumed that the location of particle beam impaction on the vaporizer does not significantly affect the thermal decomposition of vaporized molecules.

        Maximizing the particle transmission efficiency (TE) of an inlet requires careful ADL alignment so that the overlap of the particle beam with the vaporizer surface is maximized. Conventionally, an ADL is aligned manually in an iterative

process of positioning a monodisperse particle beam (typically size-selected 300 nm $NH_4NO_3$ particles) near the center of the vaporizer relying on the particle signal vs. lens movement to find the edges, which can be time-consuming and prone to human error. It is assumed that the particle beam from an ADL is well collimated across the particle diameters of interest and that the variability in particle beam center position of different particle sizes is negligible. Regular beam position measurements of monodisperse aerosols are recommended to be able to detect any changes in beam position over time, which have been

observed occasionally when shipping the instrument and/or on mobile platforms. Some AMS instruments are equipped with a capture vaporizer (CV) to fully vaporize particles (Xu et al., 2017). Compared to the standard vaporizer, more careful alignment is needed when using a CV whose cavity entrance diameter is smaller (2.5 mm OD, so 44% of the cross-sectional area of the standard vaporizer, perpendicular to the beam path).

        Manufacturing ADLs require tight mechanical tolerances, and at times, particle beam focusing and pointing are

imperfect (Williams et al., 2013). One way to monitor the beam-focusing ability of an ADL is by taking a photograph of the particle deposition pattern of polydisperse aerosols on a flat surface located in front of the vaporizer. However, this method does not provide the beam information of particles of specific diameter. A beam width probe (BWP, Aerodyne Research) has been used previously to diagnose the aerosol beam width and center position relative to the main AMS axis (Huffman et al., 2005; Salcedo et al., 2007). The BWP consists of a thin wire (typically 0.5 mm thick) that is moved in steps in front of the

skimmer upstream of the vaporizer/ionizer. AMS concentration measurements from a stable particle source are used to quantify the position-dependent signal attenuation of the particle beam by the wire, which is used to derive aerosol beam width and





position (Huffman et al., 2005). The particle beam width produced by the PM$_1$ lens for most particle types appears to be narrow enough to fully overlap the AMS vaporizer (Huffman et al., 2005). Thorough size-dependent beam width/position analysis of aerosol beams from the PM$_1$ lens, PM$_{2.5}$ lens, and HPL lens have not been reported to date. Moreover, previous BWP measurements were performed in only one dimension. Potentially, ADL imperfections may cause elliptical or irregularly shaped particle beams. To diagnose the homogeneity in particle beam width and position, BWP measurement in both dimensions orthogonal to the beam path are preferable.

While sampling aerosols on aircraft platforms, aerosols are drawn into the airplane through a dedicated inlet (*e.g.,* HIAPER Modular Inlet, HIMIL, https://www.eol.ucar.edu/content/air-sample-inlets) that uses one or several diffusers to slow down the airspeed to reduce particle losses in the sampling lines downstream. Once in the aircraft, the particle-containing airflow is delivered to the instrument at a pressure typically close to that outside the aircraft, which varies substantially with altitude. To achieve consistent particle focusing performance, the lens entry pressure needs to be kept within ~10–15% of the design value. When the lens operating pressure changes beyond that range, the particle TE changes substantially, with lower (higher) pressures favoring the TE of smaller (larger) particles (Bahreini et al., 2003; Liu et al., 2007). The first reported pressure-controlled inlet (PCI) for an AMS (Bahreini et al., 2008) maintained a constant ADL upstream pressure up to ~6.5 km altitude. That PCI consisted of a small cylindrical volume upstream of the lens between two critical orifices (COs) kept at constant pressure by a PID-controlled valve pumping the excess flow from the volume. Further improvements to this PCI design with larger COs, and a newly designed expansion volume between the CO downstream of the PCI and the ADL, enabled stable particle sampling up to 12 km altitude over the 50–750 nm $d_{va}$ range (Guo et al., 2021). As noted in Guo et al. (2021), that PCI design (CU PCI-C) is not suitable to be operated at lower input pressures (larger COs and lower PCI operating pressure) without major particle losses, and it does not benefit from the new, wider size-transmission ADLs that have been recently demonstrated (Williams et al., 2013; Xu et al., 2017) due to its own limited particle transmission. Hence a new PCI design that addresses these shortcomings while keeping residence times small is needed.

Recently, an alternative PCI was developed using a pinched o-ring as a flow restriction, which can operate at up to 20 km altitude (Molleker et al., 2020) and was tested with a PM$_{2.5}$ lens. This system may be more compact and lighter since an additional pump for the PCI excess flow is not required. However, limitations on the reproducibility of the o-ring diameter and shape can lead to significant particle losses that are hard to diagnose (Molleker et al., 2020). Clemen et al. (2020) demonstrated an improved aircraft inlet for the Aircraft-based Laser ABlation Aerosol MAss spectrometer (ALABAMA) by combining the PCI of Molleker et al. with a newly designed aerodynamic lens that significantly enhanced particle transmission (50% TE at ~3 μm $d_{va}$) compared to the aircraft inlets described above. These studies, however, did not investigate the transmission of small particles through the inlet, which is important for sampling particle growth events in both urban plumes (Allan et al., 2003b) and the upper free troposphere (Williamson et al., 2019). Sampling small particles (below ~100 nm $d_{va}$) has been a weakness of ADLs designed for large particle transmission (Williams et al., 2013; Xu et al., 2017).

Here we present a newly developed inlet system, consisting of a redesigned PCI and incorporating a PM$_{2.5}$ lens. Several new diagnostics tools were developed to more accurately characterize the inlet both during development and later as



part of the in-field quality control of inlet performance. These include a lens scanning stage that allows a quick and accurate lens alignment, a two-dimensional BWP (2D-BWP) system that measures particle beam position and width vs. particle size, and an improved calibration particle generation system for particles below 100 nm diameter. Lens scanning provides unique opportunities to investigate the variability in molecular fragmentation depending on the location of particle impaction on the

vaporizer. A 2D Gaussian aerosol beam model was developed to estimate particle loss by beam broadening and irregular pointing. We use these systems to characterize the aircraft inlet in combination with several ADLs. We characterized the performance of the entire aircraft inlet system during the Technological Innovation Into Iodine for Gulfstream V (GV) Environmental Research (TI³GER) field campaign, which reached the lower stratosphere. Finally, for a full picture of particle losses in the aircraft sampling system, we also characterized the flow field inside the HIAPER Modular Inlet (HIMIL) using a

wind tunnel facility.

## 2. Methods

### 2.1. Experimental setups for inlet characterizations in the laboratory

Monodisperse aerosols of a range of particle sizes (nominally 30–850 nm mobility diameter, $d_m$) were generated. Fig. 1 illustrates the experimental setups used for aerosol generation and sampling. Inorganic salts ($NH_4NO_3$, $(NH_4)_2SO_4$, and $NH_4I$)

dissolved in water were used to generate test and calibration aerosols of $d_m$ = 250 to 850 nm. These mobility diameters cover a range of $d_{va}$ up to ~1800 nm. Inorganic aerosols were generated with an atomizer (TSI, model 3076) and then dried with a Nafion dryer (Perma Pure, model MD-110-72). The size was selected by a Differential Mobility Analyzer (DMA, TSI, model 3081) which was operated with an impactor to minimize/eliminate the contribution of larger multiply-charged particles to the test aerosol. The DMA was calibrated using polystyrene latex spheres (PSLs). The resolution of the DMA was ~10 ($d_m/\Delta d_m$,

where $\Delta d_m$ refers to the full width at half-maximum of the distribution in log diameter space).

For smaller monodisperse particle generation ($d_m$ ~ 30–300 nm), an evaporation–condensation aerosol generator was used (Sect. S1). Oleic acid was evaporated in a heated glass bulb and then quenched with zero air to generate a monomodal particle size distribution via condensational narrowing of the distribution. An impactor was not used here to reduce the potential for multiply-charged particles. Instead, the aerosol generation system was tuned to produce relatively small particle modes, so

that the desired size to select was at the right shoulder of size distribution. Consequently, larger particles that would be transmitted as doubly-charged were minimal (see Sect. S1).

A TSI condensation particle counter (CPC model 3010, flow rate = 1 l min⁻¹) was used to measure particle number concentration. Sampling line pressure and temperature were monitored and logged continuously, so that CPC counts could be converted to number concentration at standard temperature and pressure (scm⁻³) for direct comparison to AMS-measured mass

concentrations (μg sm⁻³). The prefix "s" stands for standard temperature (273 K) and pressure (1013 hPa) conditions, per NASA convention (sometimes denoted as "μg m⁻³ STP"). When generating monodisperse oleic acid particles of < 70 nm, very high number concentrations need to be generated to achieve usable, but still modest mass concentrations, leading to saturation



of the CPC (nominally designed for up to $10^4$ particles cm$^{-3}$, although in practice some saturation can often be observed at slightly lower concentrations). To avoid this problem, a particle dilution assembly was used upstream of CPC. The assembly

consists of a short section (30 cm) of thin stainless steel tubing (ID = 0.89 mm) in parallel with a filter (Model 30/25, Balston Inc.) (Fig. 1). Most of the flow goes through the filter, which allows removing a large majority of the particles reproducibly (~96%, equivalent to ~25x dilution). The dilution factor is computed based on a 10 Hz flowmeter measurement of the filter flow and was regularly confirmed by rapid back-to-back concentration measurements (Fig. S2.1).

Polydisperse aerosols were used for the size-resolved BWP (SR-BWP) measurements (Sect. 3.3). They were

generated by nebulizing an $NH_4NO_3$ solution (typically ~0.05 M). The SR-BWP analysis requires a stable high concentration of aerosols with a broad size distribution. To increase the stability of the size distribution (in both shape and overall concentration), a mixing/buffer volume (13 L with a residence time of ~1 min) was incorporated into the setup. The AMS and a Scanning Mobility Particle Sizer (SMPS, Classifier model 3080 with CPC 3775) were used to monitor the stability of polydisperse aerosol size distribution. A Grimm optical particle counter (OPC; model 1.109) was also used to provide

supplementary size distribution measurements. A 3D-printed nebulizer, a modified, larger version (Fig. S7.1) of the one described in Rösch and Cziczo (2020) with different fittings and critical orifice mounting, was used for these experiments. The peak diameter in the volume distribution from the nebulizer was ~800 nm $d_{va}$, which greatly improved signal to noise when performing 2D-BWP analysis. The peak diameter from the TSI nebulizer was ~220 nm $d_{va}$ for the $NH_4NO_3$ solution concentrations used here (~0.05 M, Fig. S7.1). Nebulized aerosols were dried by mixing with a flow of dry zero air (dry ZA

and nebulizer flows were 12.5 and 3.5 vlpm, respectively).

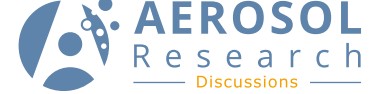

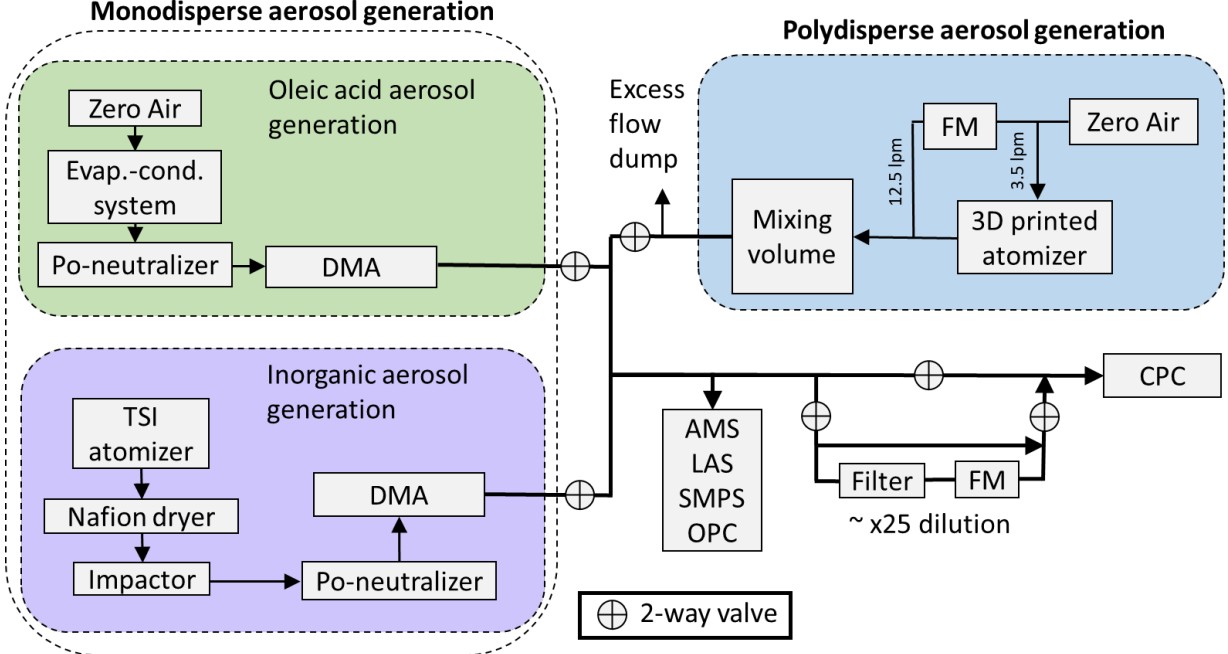

**Figure 1:** Schematic of the experimental setups for inlet characterizations in the laboratory. Monodisperse inorganic or oleic acid aerosols were used for most of the tests and calibrations. CPC particle dilution was enabled during the oleic acid particle generation below 100 nm diameter. Polydisperse aerosols (NH₄NO₃) were used for 2D-BWP experiments. FM stands for flow meter. When operating both the TSI and 3D printed atomizers, the Zero Air pressure was set to ~2.4 bar (~35 psi).

### 2.2. Aerodyne aerosol mass spectrometer (AMS)

An AMS measures the chemical composition of non-refractory aerosols. The particles are collimated by the ADL into a narrow beam, followed by flash vaporization on a 600°C porous tungsten vaporizer. Vaporized gases are subject to electron ionization, and the mass-to-charge ratio of the ions produced is measured by high-resolution time of flight mass spectrometry (DeCarlo et al., 2006; Canagaratna et al., 2007). AMS sensitivity to nitrate was calibrated with monodisperse NH₄NO₃ single particles in event trigger (ET) mode (Kimmel, 2016). Sensitivity to other species was quantified by the relative ionization efficiency (RIE) to nitrate as measured in regular MS mode (Canagaratna et al., 2007):

$$C_s = \frac{10^{12} \cdot MW_{NO_3}}{CE \cdot RIE_s \cdot IE_{NO_3} \cdot Q \cdot N_A} \cdot \sum_{all,i} I_{s,i}, \qquad (1)$$

$$RIE_s = \frac{IE_s}{MW_s} \cdot \frac{MW_{NO_3}}{IE_{NO_3}} \cdot RIE_{NO_3}, \qquad (2)$$

where $C_s$ is the mass concentration of species $s$ (µg m⁻³), $MW_{NO_3}$ and $MW_s$ are the molecular weights of nitrate and species $s$ (g mol⁻¹), $CE$ is the particle collection efficiency, $RIE_s$ is the relative (to nitrate) ionization efficiency of species $s$, $IE_{NO_3}$ is the ionization efficiency of nitrate-based on NO⁺ and NO₂⁺ only, $RIE_{NO_3}$ is the relative ionization efficiency of nitrate-based on



all fragmentation ions, $Q$ is the volume flow rate into the AMS (cm$^3$ s$^{-1}$), $N_A$ is Avogadro's number, $I_{s,i}$ is the ion signal from

ion $i$ produced from species $s$ (Hz), and the $10^{12}$ factor accounts for unit conversions.

The particle collection efficiency ($CE$) is defined as:

$$CE = E_S \cdot E_L \cdot E_b, \tag{3}$$

where $E_S$ is the shape transmission factor, which accounts for the particle loss caused by additional beam width broadening

due to non-spherical particle shapes (Huffman et al., 2005). In this study, we assume $E_S = 1$ following Huffman et al. (2005)

and Salcedo et al. (2007), who showed that $E_S \sim 1$ for ambient and typical laboratory particles. $E_L$ is the lens transmission

efficiency for spherical particles, which depends on the ADL design. The $E_L$ term includes particle losses inside the ADL, and

losses due to the particles that exit the ADL but fail to hit the vaporizer. $E_b$ is the composition and phase-dependent particle

bounce loss correction factor, due to particle bounce at the surface of the standard AMS vaporizer (Middlebrook et al., 2012).

An AMS can measure mass size distributions using the particle time-of-flight (PToF) mode (Jayne et al., 2000;

Jimenez et al., 2003; Drewnick et al., 2005). The particle $d_{va}$ is quantified by measuring the particle time-of-flight ($t_p$) between

the opening of a chopper slit and the chemical detection. PToF parameters depend on the ADL operating pressure ($P_{Lens}$).

Particle speed ($v_p$) vs. $d_{va}$ should be calibrated for each lens and operating lens pressure. The particle velocity calibration

equation (Allan et al., 2003a; Bahreini et al., 2003) is:

$$v_p = \frac{L_c}{t_p} = v_l + \frac{v_g - v_l}{1 + (d_{va}/D^*)^b}, \tag{4}$$

where $L_c$ (m) is particle flight length between the chopper and the vaporizer (0.293 m in this study), $t_p$ (s) is the measured

particle time-of-flight, $v_g$ (m s$^{-1}$) is a fitting parameter typically interpreted as the gas velocity at the exit the nozzle at the end

of the lens, $v_l$ (m s$^{-1}$) is the gas velocity within the aerodynamic lens. $D^*$ (nm) and $b$ (unitless) are additional fitting parameters.

$v_p$ as a function of $d_{va}$ is obtained by fitting the measured $v_p$ vs. $d_{va}$ points using the latter part of Eq. 4. We constrain $v_l$ (to

values from literature, Table S9.2) and $v_g$ (by measurement of the air signal delay), then $D^*$ and $b$ are fitted. $d_{va}$ of the calibration

particles is calculated from the particle volume-equivalent and mobility diameters as (DeCarlo et al., 2004):

$$d_{va} = \frac{\rho_p}{\rho_0} \frac{d_{ve}}{\chi_v} = \frac{\rho_m}{\rho_0} \cdot S \cdot d_m, \tag{5}$$

where $\rho_p$ is particle density (g cm$^{-3}$), $\rho_0$ is the density of water (1 g cm$^{-3}$), $d_{ve}$ is volume equivalent diameter (nm), $\chi_v$ is the

vacuum (*i.e.,* free-molecular regime) dynamic shape factor (= 1 for spheres and > 1 for non-spherical particles), $\rho_m$ is the

material density (g cm$^{-3}$), $S$ is the Jayne shape factor, $d_m$ is the DMA mobility diameter (nm) (Jayne et al., 2000; DeCarlo et

al., 2004). $S$ can be estimated by comparing $d_{va}$ measured by AMS in PToF mode and $d_{va}$ estimated by the latter part of Eq. 5

if the bulk density of the material is known. In this work, both regular (single-slit) and efficient PToF mode (ePToF), which

uses a multiple-slit configuration with a much higher duty cycle than the standard chopper (50% vs 2%) were used. The

parameters in Eq. 4 were calibrated for both PToF and ePToF.

In this study, the highly customized University of Colorado high-resolution time of flight aerosol mass spectrometer

(CU HR-AMS, hereinafter AMS for short) (Nault et al., 2018; Schroder et al., 2018; Guo et al., 2021) was used both in the



laboratory (without HIMIL, Fig. 2) and for field measurements during the TI³GER campaign (Sect. 2.6). For ambient aerosol measurements, the composition-dependent collection efficiency (Matthew et al., 2008; Middlebrook et al., 2012) was applied to account for the particle bouncing efficiency ($E_b$) assuming that aerosols were internally mixed, which is typical of accumulation mode aerosol in remote locations (Murphy et al., 2006). If aerosols are externally mixed (e.g. seasalt mode in

the marine boundary layer), this can cause higher uncertainty.

Ambient aerosol volume concentration (μm³ sm⁻³) was estimated from the AMS chemical composition ($V_{chem}$, Eq. 6) and compared with with the physical aerosol volume ($V_{phys}$) measured from the UHSAS (Sect. 2.6.2) measurements, assuming an internally mixed aerosol distribution.

$$V_{chem} = \left(\frac{OA}{\rho_{OA}} + \frac{SO_4 + pNO_3 + NH_4}{1.75 \, g \, cm^{-3}} + \frac{Cl}{1.52 \, g \, cm^{-3}} + \frac{Seasalt}{1.45 \, g \, cm^{-3}}\right) \cdot 10^{-6} \tag{6}$$

Aerosol chemical compositions (μg sm⁻³) were measured by AMS. Seasalt density (1.45 g cm⁻³) was taken from Guo et al. (2021) assuming partially deliquesced particles (Brock et al., 2019). OA density ($\rho_{OA}$, g cm⁻³) was estimated from O/C and H/C ratios (Kuwata et al., 2012). O/C and H/C ratios were calculated using improved ambient elemental analysis (Canagaratna et al., 2015). 10⁻⁶ is the unit conversion factor. During TI³GER, $V_{chem}$ was estimated without rBC (refractory black carbon) since that measurement was not available. The contribution of rBC to PM₁ particle mass is low, typically below ~2% in the

Northern Hemisphere during ATom campaigns (Hodzic et al., 2020; Brock et al., 2021).

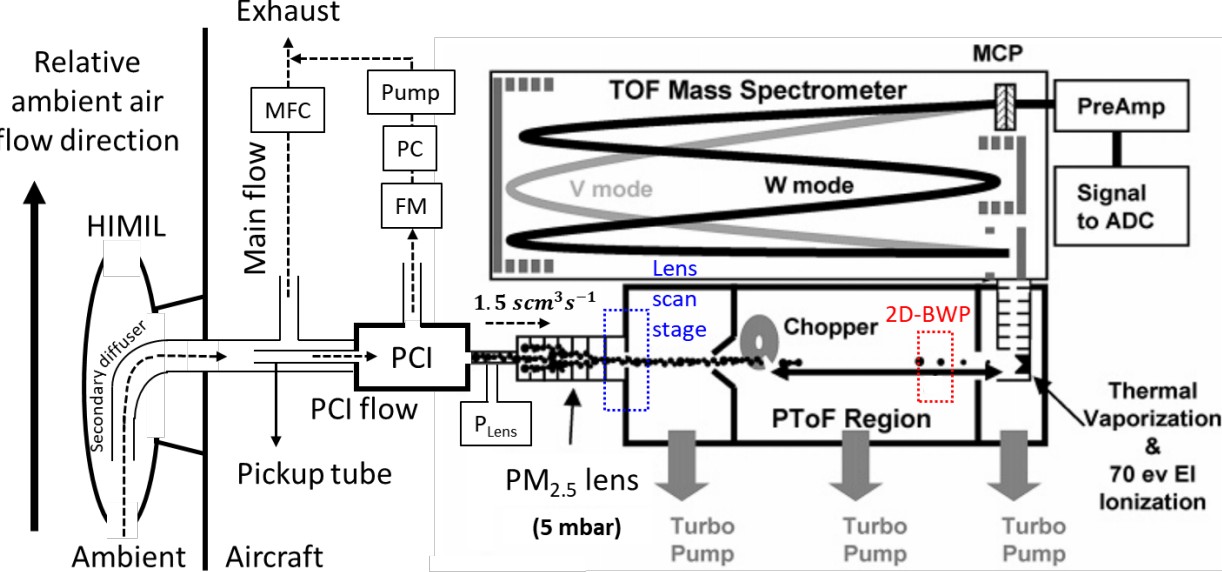

**Figure 2: Simplified schematic diagram of the CU-HR-AMS setup for aircraft sampling. The AMS schematic is from DeCarlo et al. (2006). Ambient air is drawn into the aircraft through the HIMIL inlet (Fig. S20.1). Part of the air is drawn into the PCI before the**
**AMS. Additional air is exhausted to reduce inlet residence time. PCI pressure ($P_{PCI}$) is measured and controlled by a pressure controller (PC) and the flow through PC is monitored by a flow meter (FM). In the AMS, aerosols are collimated into a narrow**



beam followed by vaporization and ionization for mass spectrometry. More detailed schematic diagrams for the lens scan stage and 2D-BWP unit can be found in **Fig. 3**.

## 2.3. New tools for lens alignment and aerosol beam diagnostics

### 2.3.1. 2D Lens scan imaging stage

We have developed a new, fully automated ADL alignment stage that provides fast, accurate, and reproducible lens alignment. The original alignment stage (that is installed on standard commercial AMSs) was replaced with a custom-built 2D lens scan stage that consists of a linear XY stage (Thorlabs Mod XYT1) with stronger, custom springs and a new vacuum interface (Fig. 3 and Sect. S5). The manual actuators were replaced with electronic stepper motor linear actuators (Thorlabs Inc., model

ZFS13B) in the two orthogonal directions to the particle beam (X and Y axes). Note that the X-Y axes of the lens scanning stage are offset 15 degrees from the X-Y axes of the BWP due to mechanical limitations. For the vacuum interface, a custom edge-welded bellows tube provides both secure vacuum sealing and a sufficient range of motion for the ADL (about 5 mm). The electronic actuators are computer-controlled and can do a full 2-dimensional scan of the vaporizer in ~25 minutes. The precision and reproducibility of the electronic actuators is < 50 μm, which translates to a positional reproducibility of better

than 125 μm when projected on the AMS vaporizer.

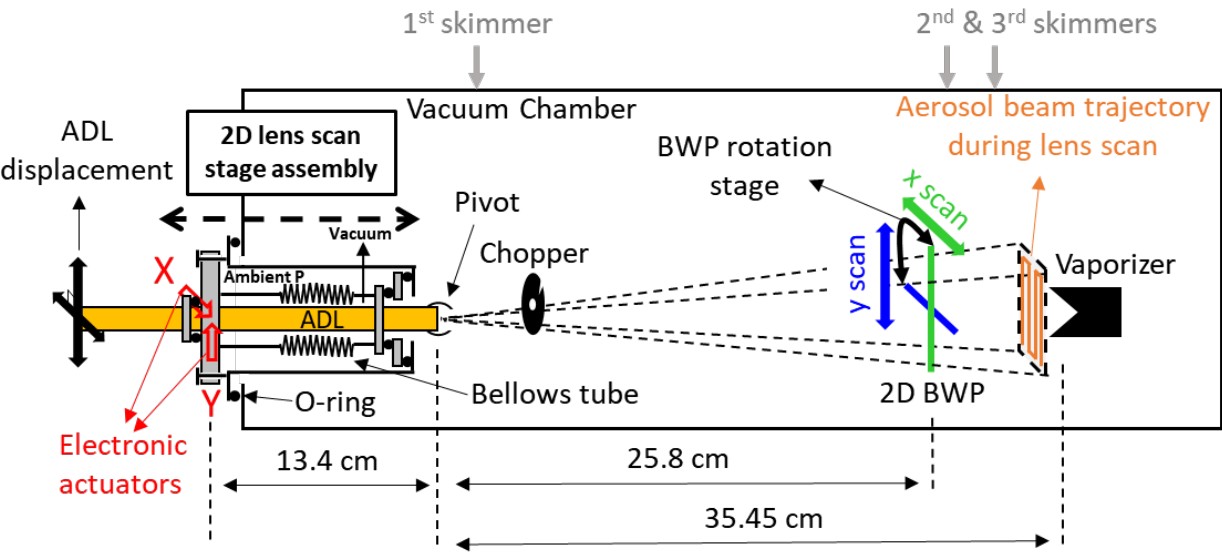

**Figure 3: Schematic of lens scan stage and its scanning area. This diagram is not to scale and the range of lens scanning angles is exaggerated here for clarity. During the lens scanning process, the chopper is in the continuously open position and the BWP is**
**positioned outside the lens scanning range. Details of the 2D BWP system are described in Sect. 2.3.2. The lens scan X-Y coordinate is offset 15 degrees from the BWP X-Y coordinate (Fig. S6.1-2).**





### 2.3.2. Size resolved beam width probe in two dimensions (2D-SR-BWP)

The BWP provides information on the particle beam position and width (Jayne et al., 2000; Huffman et al., 2005). Previously reported BWP analyses derived the particle beam width and beam center positions of monodisperse particles along one axis

(hereafter 1D-BWP). 1D-BWP has been used to diagnose beam broadening due to particle morphology ($E_S$) (Jayne et al., 2000; Huffman et al., 2005; Salcedo et al., 2007; Docherty et al., 2013; Willis et al., 2014). The 1D-BWP was also used to monitor the pointing and focusing stability of an ADL inlet on aircraft campaigns (Guo et al., 2021). Hence, previous efforts made to model the aerosol beam (beam position and width as a function of particle sizes) using 1D BWP assumed a symmetric Gaussian beam model since the beam profile measurements were available along only one axis (Huffman et al., 2005).

A rotational stage was built (Fig. 3, Sect. S6) for the Aerodyne BWP assembly so that BWP scans can be performed along two perpendicular directions (2D-BWP). 2D-BWP measurement of monodisperse aerosols provides both beam position and width in a 2-dimensional plane. The rotation stage consists of a high torque servo (Bilda Mod 2000) controlled by a USB servo controller (Polulu Micro-Maestro 6-ch USB controller) and a custom-built gear drive that allows up to 135-degree turn of the BWP unit.

Additionally, instead of recording total mass signals during 2D-BWP operation, particle size distributions can be recorded continuously in AMS PToF mode. This way, one can obtain size-resolved (SR) particle beam information along the two orthogonal axes (2D-SR-BWP) with one set of measurements. PToF measurements were carried out for 9 seconds at each BWP position. For aerosol input, stable polydisperse $NH_4NO_3$ particle distributions were generated (see Sect. 2.1) with a typical modal diameter of $d_{va} \sim 800$ nm.

Typically, the BWP is scanned in 0.1 (or 0.05) mm intervals with a total of 27 (or 54) steps on each axis. Systematic scan-to-scan offsets on BWP position (due to hysteresis in the stepper motor and normally within ± 0.2 mm) can be identified using the BWP airbeam position and corrected post-acquisition (Fig. S16.2). The BWP positions that correspond to the vaporizer center were measured for both axes and the offsets were applied to our data (Sect. S6) to put both the lens and BWP scan on a consistent coordinate system. For visualization, aerosol attenuation factors (Eq. 7) are constructed as a function of

BWP position and $d_{va}$. Then the attenuation factors vs. BWP position at a given particle size are fitted with Gaussian curves and normalized giving the normalized attenuation factor ($A$).

$$A = (S_{ref} - S_{block})/S_{ref}, \tag{7}$$

where $S_{ref}$ is the aerosol signal when BWP is not blocking the beam and $S_{block}$ is the aerosol signal when BWP is blocking partially (or completely) the aerosol beam at a certain BWP position.

### 2.3.3. Particle beam profile model

A model was developed to estimate the fraction of particles existing from the ADL that impact the vaporizer. Following the methods of Huffman et al. (2005), the model simulates the particle beam, assuming a 2-D Gaussian distribution, and is fitted to the results of 2D-SR-BWP measurements. The measured beam width ($\sigma_{IDG}$, following the convention from Huffman et al.,





2005) of the particle beam is defined as one standard deviation from the Gaussian fitting of signal attenuation along the BWP
wire positions. When the Gaussian fitting fails due to high noise for a given $d_{va}$ size bin, the raw data is median-smoothed by
two steps to up to a quarter of the original BWP positional resolution, and then the Gaussian fit is performed.

$\sigma_{IDG}$ differs from the actual particle beam width due to the convolution effect with the BWP wire width (0.5 mm). In
the model, the relationship between Gaussian beam width without the wire effect ($\sigma_M$) and the width reported by BWP
measurement ($\sigma_{IDG}$) is found by simulating the same wire movement as in the measurements (Fig. 4). When the beam width
is narrow (< 0.2 mm $\sigma_M$), BWP measurement reports ~ 0.2 mm $\sigma_{IDG}$ due to wire effect. Over 0.2–0.6 mm $\sigma_M$ range, $\sigma_{IDG}$ does
not exhibit significant bias. For $\sigma_M$ larger than 0.6 mm, $\sigma_{IDG}$ is biased low since a fraction of particles is not captured by the
vaporizer.

Since the beam width measurements by BWP are done upstream of the vaporizer, an additional linear correction is
necessary to account for particle beam width broadening between the BWP plane and the vaporizer surface. Given the constant
radial speed of aerosol particles in the vacuum chamber,

$$\sigma_v = \frac{L_{NV}}{L_{NB}} \, \sigma_M, \tag{8}$$

where $\sigma_v$ is the particle beam width at the vaporizer plane, $L_{NV}$ (length from lens exit nozzle to the vaporizer) and $L_{NB}$ (length
from lens exit nozzle to BWP) were 35.45 cm and 25.8 cm, respectively (Fig. 3). The apertures of the three skimmers ($\Omega =$
$5.17\text{x}10^{-4}$ sr, $1.39\text{x}10^{-4}$ sr, $1.17\text{x}10^{-4}$ sr for the first, second, and third skimmers) between ADL and vaporizer have a larger
beam angle than that of the vaporizer ($\Omega = 0.81\text{x}10^{-4}$ sr), thus those skimmers should not affect BWP measurements as long
as they are properly aligned. In this study, the vaporizer appears to be misaligned that the vaporizer was partially blocked by
the third skimmer (Fig. S5.8) which should not significantly affect the beam width measurements and modeling.

After obtaining the particle beam width ($\sigma_v$) along both perpendicular axes as a function of $d_{va}$, we used a general 2D
Gaussian probability density function to simulate the beam profile. This allows the modeling of beams with elliptical cross-
sections:

$$g_{2D} = \frac{1}{2\pi\sigma_x\sigma_y} exp(-(\frac{(x-x_0)^2}{2\sigma_x^2} + \frac{(y-y_0)^2}{2\sigma_y^2})), \tag{9}$$

where $\sigma_x$ and $\sigma_y$ are $\sigma_v$ in the x and y directions, $x_0$ and $y_0$ are the beam center positions in the x- and y-axes relative to the center
of the vaporizer, $d_v$ is diameter of the vaporizer. In the model, transmission efficiency ($TE_{mod}$ accounting for transmission
between ADL and vaporizer) is calculated as the fraction of the integrated 2-D Gaussian function inside the vaporizer
perimeter. Note that the modeled transmission efficiency does not account for particle losses inside and upstream of ADL. The
main difference from Huffman et al. (2005) is that the BWP measurements are available for two perpendicular axes in this
work. Thus, the assumption used by Huffman et al. (2005) that the beam cross-section is circular is not needed here. Also,
unlike Huffman et al. (2005) where $x_0$ and $y_0$ are assumed to be zero, we measured these parameters as a function of $d_{va}$. The
model is used to diagnose the beam pointing/focusing characteristics of a given lens as a function of $d_{va}$ (Sect. 3.2.3) and to
optimize the alignment of the lens so that $E_L$ is maximized across the widest possible range of particle sizes (Sect. 3.3.2).



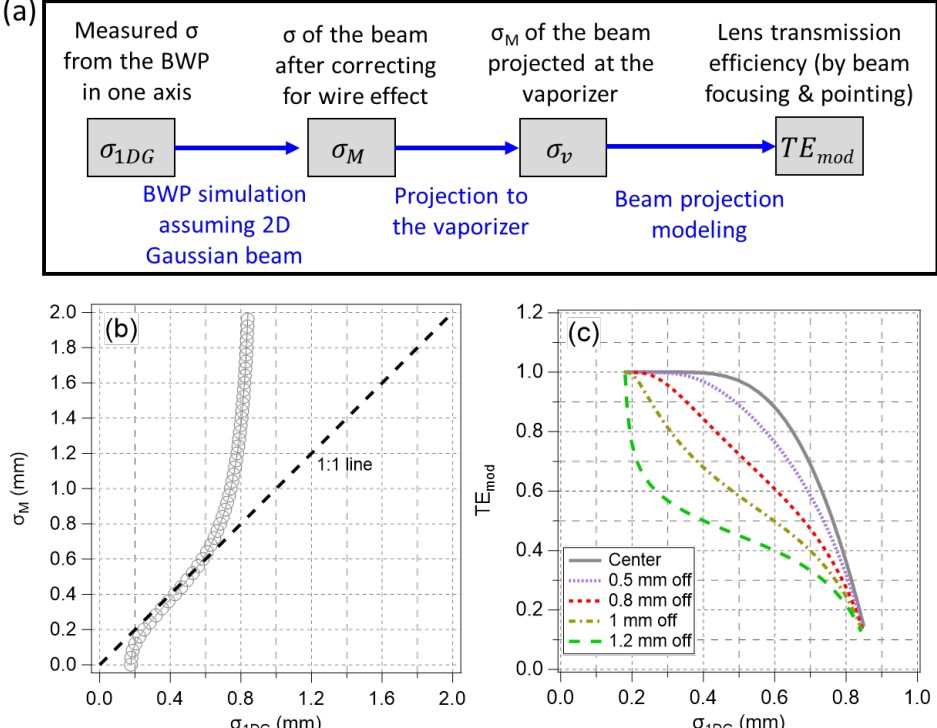

**Figure 4: Particle transmission modeling procedure with input from 2D-SR-BWP measurements. (a) Beam width conversion flow chart for TE estimation as a function of $\sigma_{1DG}$. (b) Modeled Gaussian beam width after correcting for the beam broadening due to BWP wire thickness ($\sigma_G$) vs measured width by BWP ($\sigma_{1DG}$) (c) Modeled TE vs. $\sigma_{1DG}$ at different beam center positions relative to the center of the vaporizer. In these examples, a perfectly symmetric circular Gaussian beam is assumed.**

## 2.4. Measurement of particle transmission efficiency (TE)

For laboratory measurements, overall particle TE of a given inlet system using monodisperse particles was calculated by comparing AMS-based mass ($AMS_{mass}$) to CPC-based mass ($CPC_{mass}$) following Eq. 10–11:

$$CPC_{mass}\ (\mu g\ sm^{-3}) = 10^{-9} \cdot N_{CPC}\ (scm^{-3}) \cdot \frac{\pi}{6} \cdot d_m(nm)^3 \cdot \rho_m \cdot S \tag{10}$$

$$TE = \frac{AMS_{mass}\ (\mu g\ sm^{-3})}{CPC_{mass}\ (\mu g\ sm^{-3})} \tag{11}$$

where $N_{CPC}$ is the particle number concentration measured by CPC and the term $\rho_m \cdot S$ in Eq. 10 stands for the effective particle density (g cm⁻³) (DeCarlo et al., 2004). The $S$ for laboratory-generated dry $NH_4NO_3$ is 0.8 (Jayne et al., 2000) and $S$ for dried $NH_4I$ was 0.83 ± 0.07 (Table. S4.1). As noted in Sect. 2.1, oleic acid was used for TE measurements below 300 nm $d_{va}$ and inorganic compounds ($NH_4NO_3$, $(NH_4)_2SO_4$, $NH_4I$) were used for larger diameters. Data was typically averaged for 5 min per TE point. The 1 min acquisition sequence used in this work for the AMS consists of 6 s chopper closed, 46 s chopper open, and 8 s ePToF cycles, as typically used on aircraft campaigns (Nault et al., 2018).



On aircraft platforms, CE from Eq. 3 can be expanded, accounting for the particle transmission of the aircraft inlet ($E_I$), of the tubing between the aircraft inlet and of the PCI ($E_T$), and PCI ($E_{PCI}$), as:

$$CE_{\text{total}} = E_S \cdot E_L \cdot E_b \cdot E_{PCI} \cdot E_T \cdot E_I \qquad (12)$$

$E_T$ for our aircraft plumbing configuration was estimated from sample flows and tubing dimensions and shapes (Guo et al., 2021; Bourgeois et al., 2022) (Sect. S18). $E_I$ is the transmission efficiency as particles enter the HIMIL which is close to unity (Stith et al., 2009). The measured $TE$ of the inlet system in the laboratory after correcting for $E_b$ is hence the product of the TEs of the inlet components that particles travel through during the measurement:

$$TE = E_L \cdot E_{PCI} \cdot E_T, \qquad (13)$$

$E_T$ in the laboratory plumbing configuration was estimated to be close to 1 by comparing two particle counters upstream and downstream of the plumbing line between AMS and CPC (see Sect. S3). When measuring TE without PCI installed, $E_{PCI} = 1$. By comparing $d_{va}$ calculated from Eq. 5 vs. the AMS measurements, potential evaporation and the presence of doubly charged particles were routinely monitored.

When measuring the TE of standalone ADLs, PM$_1$ lens, PM$_{2.5}$ lens (S/N = 66), and HPL (S/N = 12) (Sect. 3.1), TE is equivalent to $E_L$. A critical orifice (CO) with a 120 μm diameter was installed upstream of the system as it resulted in the optimal lens entry pressure at Boulder, Colorado altitude (~0.8 atm), where all the laboratory experiments were conducted. Hereafter, a critical orifice in a standard AMS configuration (*e.g.,* without PCI) will be referred to as CO$_{std}$. Orifices used in this study are made of platinum, 6.35 x 0.125 mm dimension (diameter x thickness), and the aperture is conically drilled (PerkinElmer Inc.). An expansion volume (EV) version C (EV-C, Fig. 5a) was used as a relaxation chamber (Wang and McMurry, 2007) between the CO$_{std}$ and ADL to minimize potential particle losses after the supersonic expansion at the CO$_{std}$. An expansion volume is not part of the standard AMS configuration.

**2.5. Pressure-Controlled Inlet (PCI) designs tested**

A constant pressure in the ADL ensures consistent aerodynamic focusing of the aerosol onto the AMS vaporizer (Zhang et al., 2004b). Without active pressure control, variable ambient pressure as altitude changes leads to changes in lens pressure during aircraft deployments. This results in a change in size calibration parameters, lens transmission efficiency vs. size, and air flow rate into the AMS (Bahreini et al., 2003). A PCI is a device to maintain constant pressure upstream of the aerodynamic lens during flights. Two critical orifices are used, the first critical orifice at the inlet of the PCI (CO$_{up}$) operating between ambient pressure and the intermediate-pressure volume (IPV), and the second critical orifice (CO$_{down}$) at the IPV exit. Note that CO$_{down}$ replaces CO$_{std}$ in the standard AMS configuration. The IPV is pumped by a vacuum pump (Vacuubrand MD1 in this work). The pressure at the IPV is referred to as $P_{PCI}$. The pump flow is controlled by a pressure controller (Mod PC3P Alicat Scientific Inc.), which keeps $P_{PCI}$ constant. CO$_{up}$ must be large enough to ensure enough excess flow (besides the inlet flow strictly needed by the AMS) at all altitudes; a larger CO$_{up}$ will shorten the overall residence time, so ultimately the limitation is the pumping capacity of the pump at $P_{PCI}$. $P_{PCI}$ needs to be maintained below the lowest inlet line pressure (accounting for effects



of ram pressure, inlet line pressure drops, and — for the NASA DC-8 installation — lower over the wing pressure) that will be sampled, in order to always be able to draw air into the PCI system. $CO_{down}$ size is chosen to maintain the required AMS flow and lens pressure, given a constant $P_{PCI}$. However, lower $P_{PCI}$ is more prone to the loss of large particles during and after expansion (Chen et al., 2007; Guo et al., 2021) due to impaction on either the backside of $CO_{down}$ or the wall downstream of the $CO_{down}$. Thus, when using the larger $CO_{down}$ needed for lower $P_{PCI}$, careful design of the downstream EV is key to minimizing particle losses.

The performance of the previous PCI designs (CU PCI-A, B, C) used as part of the CU-HR-AMS during NASA airborne missions are further described in Sect. S11. In this section, we compare the two most recent University of Colorado PCI designs used for the ATom and FIREX-AQ campaigns (Fig. 5a, CU PCI-C) with the newly designed PCI in this work (Fig. 5b, CU PCI-D). Hereafter, all PCIs are referred to without CU for brevity. PCI-C has a single EV (EV-C) downstream of $CO_{down}$. When operated at lower $P_{PCI}$, TE of the PCI-C was significantly reduced (Sect. 3.4.2). To minimize particle losses at lower $P_{PCI}$, PCI-D was designed with two EVs downstream of $CO_{up}$ and $CO_{down}$ (EV-$D_{up}$ and EV-$D_{down}$). The EV-$D_{down}$ was newly designed with a conical shape motivated by Hwang et al. (2015) to minimize air recirculation and thus minimize particle loss after the supersonic expansion at $CO_{down}$. The chosen cone angle and dimensions of $EV_{down}$ were informed by computational fluid dynamic (CFD) modeling. The particle transmission of the PCI was then found to be limited by the particle losses at the $CO_{up}$. The EV-$D_{up}$ was empirically designed to provide a particle relaxation volume after $CO_{up}$ and reduce particle losses. In the laboratory, PCI-C and PCI-D were further tested with several orifice sizes (besides the nominal COs in Table 1) and $P_{PCI}$ (Sect. 3.4.2).

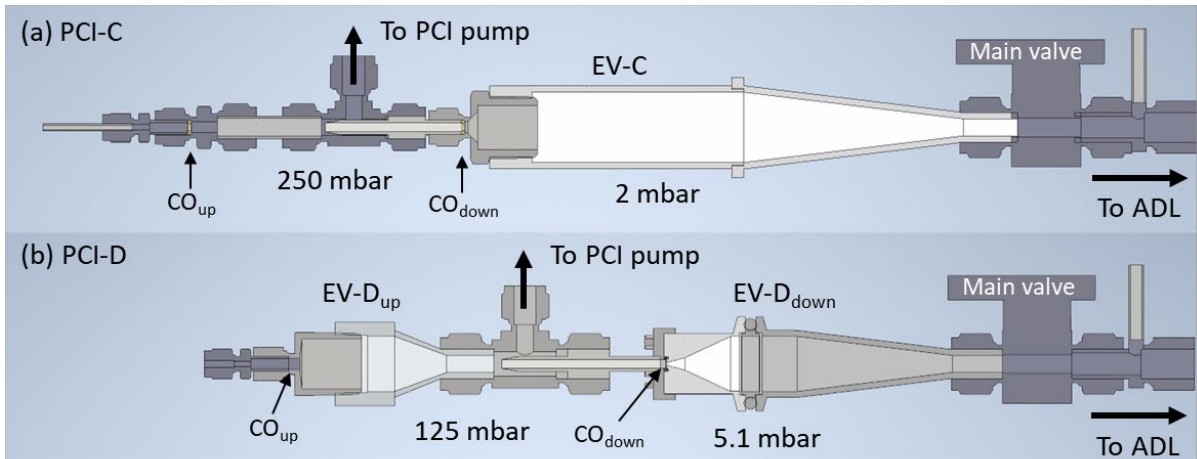

**Figure 5: 3D Model cross sections of the PCI designs tested in this work. (a) CU PCI-C (Guo et al., 2021) and (b) CU PCI-D (this work). The pressure in the IPV (between $CO_{up}$ and $CO_{down}$) is actively controlled to be constant during flights. CO diameters are shown in Table 1. The $P_{PCI}$ for CU PCI-D shown in the figure is when $d_{CO,down}$ = 300 μm.**



**Table 1: Nominal components and operating conditions of the CU PCI-C and CU PCI-D during field deployments. $d_{CO,up/down}$ refers to the diameter of the critical orifice up/downstream of the IPV.**

| PCI design | CU PCI-C (Guo et al., 2021) | CU PCI-D (This work) | |
|---|---|---|---|
| ADL used | PM$_1$ lens | PM$_{2.5}$ lens | |
| Field campaigns | ATom 1-4*, FIREX-AQ | TI$^3$GER | |
| $P_{Lens}$ (mbar) | 2 | 5.1 | |
| $d_{CO,up}$ / $d_{CO,down}$ (μm) | 350 / 220 | 450 / 350 | 450 / 300[†] |
| Set $P_{PCI}$ (mbar) | 250 | 96 | 122.6 |

*CU PCI-C was used for the later part of the ATom-1 campaign. For the earlier part, the PCI design was the same as the KORUS-AQ campaign (CU PCI-B). More detailed comparisons of previous CU PCIs can be found in Sect. S11.
[†]During NASA ASIA-AQ 2024 campaign, CU PCI-D with 400/300 μm ($d_{CO,up}$ / $d_{CO,down}$, $P_{PCI}$ = 122.6 mbar) was used and a PM$_{2.5}$ lens with improved nozzle design was deployed ($P_{Lens}$ = 5.6 mbar).

## 2.6. Airborne aerosol measurement during the TI$^3$GER field study

The Technological Innovation Into Iodine for Gulfstream V (GV) Environmental Research (TI$^3$GER) field campaign focused on technical advancements for airborne in-situ measurements up to the lower stratosphere (Yang et al., 2024a, 2024b). The TI$^3$GER campaign was conducted over Colorado (2 flights) and the Northern Pacific Ocean (6 flights) onboard the NSF/NCAR GV aircraft, on 2–29 April 2022 (https://www.eol.ucar.edu/field_projects/ti3ger). Flight altitude ranged from sea level to 14 km and latitude ranged 3–60 N. Aerosol measurements relevant to this study are the CU aircraft AMS for accumulation mode aerosol chemical composition, the NCAR Ultra-High Sensitivity Aerosol Spectrometer (UHSAS) for submicron particle size distribution (in-cabin), and the NCAR Cloud Droplet Probe (CDP) for supermicron aerosols/cloud droplet measurement (under the wing). Ancillary data including temperature, pressure, and wind speed were provided by NCAR using standard sensors on the GV aircraft.

### 2.6.1 CU Aircraft HR-ToF-AMS

A general description of AMS can be found in Sect. 2.2. During the TI$^3$GER campaign, absolute sensitivity calibrations (Sect. S15) and inlet diagnostics for the AMS were performed after each flight as well as between flights when possible (Sect. S16). For AMS particle sampling, as shown in Fig. 2, ambient air was drawn into the airplane (3–14 std l min$^{-1}$, actively controlled, depending on aircraft altitude) through an NCAR HIAPER Modular Inlet (HIMIL) (Stith et al., 2009). The HIMIL inlet used during the TI$^3$GER campaign is the "tall version" (12" tall). HIMIL was located at the left side ceiling of the GV, ~10 m behind the nose of GV (Fig. S20.1). See Sect. S20 for more details on HIMIL. The main flow into the inlet was controlled by the main mass flow controller (MFC), the flow rate of which was adjusted to maintain near isokinetic sampling at the secondary diffuser (Sect. S20). The flow toward the PCI goes through the pickup tube (Fig. 2). PCI-D was used to maintain a constant pressure upstream of the PM$_{2.5}$ lens ($P_{lens}$ = 5.1 mbar) allowing for a constant flow (1.5 scm$^3$ s$^{-1}$) into the lens. The extra PCI flow is dumped into the exhaust by the GV venturi system. More details on PCI operation can be found in Sect 2.4. The plumbing line



from HIMIL to the pickup tube is 6.35 mm OD (4.57 mm ID) stainless steel tubing with a total length of 142 cm and a cumulative bending angle of 335 deg. The tubing from the pickup tube to the PCI is ⅛" stainless steel (2.13 mm ID) with a total of 34 cm and 270 deg. cumulative turn.

### 2.6.2 NCAR UHSAS

The Ultra-High Sensitivity Aerosol Spectrometer (UHSAS, Droplet Measurement Technologies, Longmont, CO) illuminates particles with an intracavity laser (1054 nm) and relates the single-particle light scattering intensity and count rate measured over a wide solid angle (33–147°) to the size-dependent particle concentration (Kupc et al., 2018; Moore et al., 2021). During TI³GER, an in-cabin UHSAS was operated by the National Center for Atmospheric Research (NCAR). The UHSAS pulled 10 vlpm flow through a HIMIL (located on the plane belly side of GV) inlet outfitted with a stainless steel tubing line with a

90° turn inside the HIMIL followed by 0.48 cm ID conductive flexible silicone tubing (TSI) having a total of 697° of cumulative turns. Particle number concentrations between ~55 nm and ~1000 nm optical diameters ($d_{opt}$) were reported in this work.

Each UHSAS bin of scattered light intensity can be converted to particle size, based on the real part of the refractive index ($RI$) of the dry polystyrene latex spheres (PSLs) used for its calibration ($nD$=1.595). Ambient particles have a different

$RI$ than PSLs causing either an over or underestimation in particle sizes. The largest uncertainty arises in the estimation of the actual refractive index of ambient particles, discussed below (Brock et al., 2011). Then the optical diameter is calculated from the measured light scattering following the Mie theory. Total volume can be estimated from the size and particle number concentration assuming spherical particles.

We have corrected the $d_{opt}$ from UHSAS using ambient $RI$ estimated from aerosol chemical composition measured

by the AMS. RI values of 1.527, 1.554, and 1.64, were used for dry $(NH_4)_2SO_4$, $NH_4NO_3$, and $NH_4Cl$ components, respectively (Brock et al., 2021). We used 1.52 for the organic component RI following Aldhaif et al. (2018).

The effective real part RI ($\bar{n}$) of ambient aerosols was calculated as the volume-weighted mean real part RI of each component ($n$) (Sokolik and Toon, 1999; Aldhaif et al., 2018).

$$\underline{\bar{n}} = \sum_i \o_i n_i, \tag{14}$$

$$\o_i = \frac{c_i}{\rho_i} / \sum_k \frac{c_k}{\rho_k}, \tag{15}$$

where ø, $c$, and $\rho$ refer to the volume fraction, mass concentration, and bulk density of the chemical component. Chemical components were excluded from Eq. 14 and Eq. 15 when the concentrations were below their detection limits. Then each UHSAS $d_{opt}$ bin is updated with the estimated $\bar{n}$ based on Mie scattering code (Jimenez Group GitLab, 2024) which originates from Bohren and Huffman (1998). When applying the estimated $\bar{n}$ to UHSAS $d_{opt}$ size bins, it is assumed that the measured

ambient particles were internally mixed, and the contribution of BC is negligible. This assumption can cause additional



uncertainty in the marine boundary layer (MBL) where seaspray aerosols are typically externally mixed with accumulation mode aerosols.

## 3. Results and Discussion

### 3.1. Lens scan imaging of the AMS vaporizer

**3.1.1. Lens scanning procedure and particle beam width measurement**

A typical lens scan and its processed outputs are shown in Fig. 6. The details of the lens scan stage system are described in Sect. 2.3.1. A typical lens scan takes ~25 minutes allowing for faster, more accurate, and reproducible lens alignment compared to the traditional manual alignment process. In this example, 350 nm $d_m$ monodisperse $NH_4NO_3$ particles were used to locate the perimeter and the center position of the vaporizer with a $PM_1$ lens. To correct for the potential variations in aerosol source

during lens scans, signals were normalized to CPC particle counts (Fig. 6a). In Fig. 6c, aerosol signals were mapped inside the vaporizer perimeter. The diameter of the perimeter has an effective diameter of 3.6 mm, as measured from the lens scan. This diameter is smaller than the physical diameter (3.8 mm) likely due to misalignment of vaporizer position and some blockage by the third skimmer (Fig. S5.8). The resolution of the vaporizer imaging is limited by the scanning path setup and the width of the particle beam. The latter is why a lens scan with a $PM_1$ lens is typically sharper than with a $PM_{2.5}$ lens, see next section.

In this study, we found that the aerosol beam center position can depend on the particle size (Fig. 16). In such cases, we used this method to align the lens to the position where the overall lens transmission efficiency is maximized. Alignment was refined subsequently using information from the 2D-BWP (Sect. 3.2.1).

Besides the accurate location of the center and perimeter of the vaporizer, lens scanning provides information on beam width. As the particle beam is pointed towards the vaporizer edge, the aerosol signal is attenuated due to partial impaction

outside the vaporizer. The narrower the beam width, the sharper the signal attenuation on the edge of the vaporizer. In this case, an even sharper edge is provided by the third skimmer (Fig. S5.8). As an alternative to the thin wire type BWP, Huffman et al. (2005) evaluated a "Knife-Edge" type BWP, *i.e.* a flat plate moved sequentially through the beam to block an aerosol beam which was demonstrated previously in Liu et al. (1995a). In this case, the aerosol beam pointing outside of the vaporizer is the same process as being blocked by the "Knife-Edge" type BWP as long as the beam width is significantly narrower than

the vaporizer diameter. Assuming the "Knife-Edge" model, $\sigma_{IDG}$ can be obtained by fitting the attenuation curve with a sigmoidal function, assuming a Gaussian beam (Fig. S5.7). Similarly, Clemen et al. (2020), measured particle beam width by two vertical aligned detection lasers in the ALABAMA instrument. They moved the particle beam outside of the detection region by tilting the ADL and calculated the beam width based on the signal attenuation and the laser beam width.

The inset in Fig. 6a illustrates an example of signal attenuation as a function of Y actuator position. In this example,

applying the "Knife-Edge" method, the $\sigma_{IDG}$ in the BWP plane is 0.035 mm (Fig. 6a and Fig. S5.7a-b). The advantage of this technique compared to using the BWP is that the measurable beam width is not limited by the width of the wire itself (Fig. 4)

allowing finer beam width measurement, mostly limited by the resolution of the lens scan (about 0.13 mm in the vaporizer plane). However, when the beam width is too broad (> ~0.25 mm in BWP plane), the outer side of the vaporizer edge (or third skimmer perimeter) does not provide an ideal flat plate underestimating the beam width (Fig. S5.7c-d). In combination with the BWP measurements which cannot quantify widths below ~0.2 mm (in the BWP plane), lens scans can provide supplemental measurements of the beam width for smaller beam sizes (below ~0.25 mm; Fig. 11).

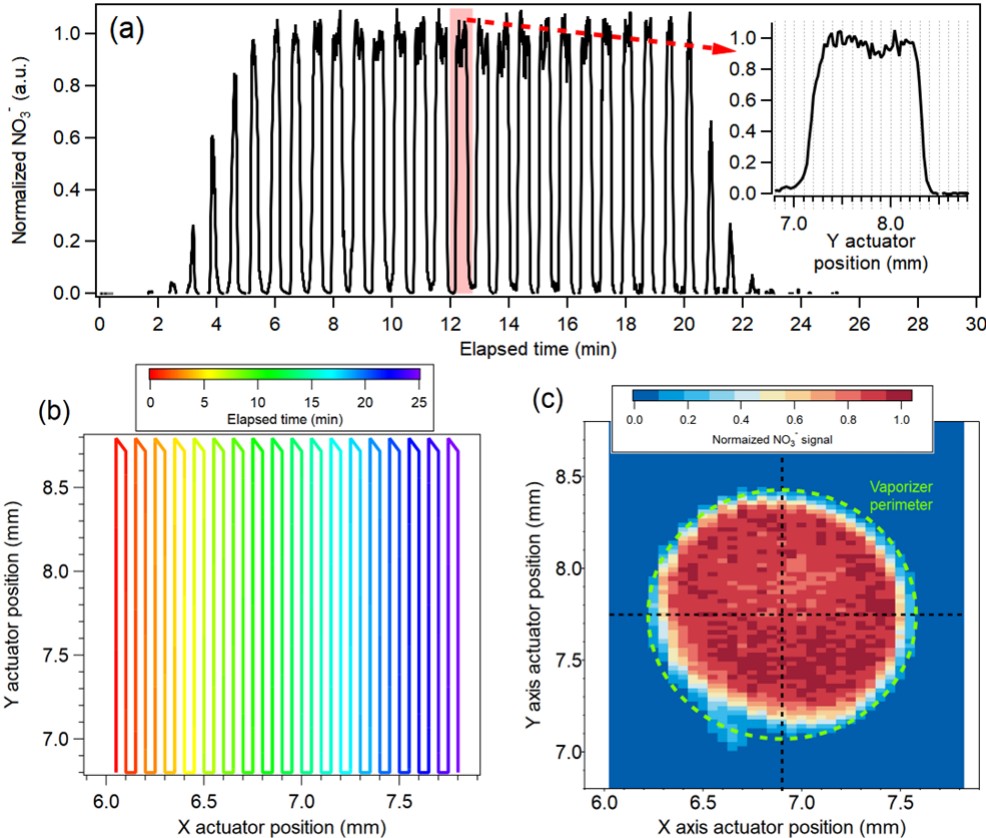

**Figure 6: (a) Time series of the AMS nitrate signal for a lens scan with 350 nm *dₘ* NH₄NO₃ particles with a PM₁ lens. The particle signal was normalized by CPC counts to correct for any variations of particle number during the scan. (b) The trajectory of X and Y actuator sweeps are colored by time. For the top inset magnifies the data in (a) for a short period and the time axis was converted to the Y actuator position during one sweep where the X axis actuator was fixed pointing to the center of the vaporizer. (c) 2D lens scan image of the vaporizer colored by normalized particle signal. The green dotted line represents the perimeter of the vaporizer projected to the actuator plane. For this scan, 1 pixel is 126 x 63 μm (X x Y axes) resolution on the vaporizer plane.**

### 3.1.2. Position-dependent decomposition on a standard vaporizer

On the surface of the vaporizer, the thermal decomposition of $NH_4NO_3$ particles produces a mixture of gases including $NH_3(g)$, $HNO_3(g)$, $NO_2(g)$, $NO(g)$, and $H_2O(g)$ (Drewnick et al., 2015). Electron ionization of $HNO_3(g)$ and $NO_2(g)$ results (after some additional fragmentation) in $NO_2^+$ and $NO^+$ which are the main ions detected by the AMS from aerosol nitrate (Canagaratna et al., 2007; Hu et al., 2017b). $NO(g)$ produces only $NO^+$ ions. Thus, for higher vaporizer temperatures leading to higher



thermal decomposition of $NH_4NO_3$, the $NO_x^+$ ratio ($NO_2^+$ to $NO^+$ signal ratio) measured by the AMS decreases. In this section, we used the lens scan technique to investigate the position-dependent decomposition patterns on the standard vaporizer for $NH_4NO_3$ and oleic acid. Fig. 7 shows lens scan images of standard vaporizers using monodisperse $NH_4NO_3$ particles with $PM_1$ and $PM_{2.5}$ lenses as well as the HPL colored by $NO_x^+$ ratio.

The results with the $PM_1$ lens and HPL show a spot near the nominal center of the vaporizer where the $NO_x^+$ ratio is
lower, indicating higher fragmentation of nitrate. There are two explanations for higher degrees of thermal decomposition of $NH_4NO_3$ (including also higher $H_2O^+$ fraction, discussed below) at the center of the vaporizer. When particles impact the center of the vaporizer, the gas-phase molecules have a higher chance of another collision with the hot vaporizer surface (and thermal decomposition) before ionization ("geometry effect"). Also, potentially, the vaporizer temperature ($T_v$) on the center of the vaporizer is slightly higher, causing higher thermal decomposition ("temperature effect"). Hu et al. (2017b) showed that the
$NO_x^+$ ratio decreased (~25–30%) as the $T_v$ setting increased from 200°C to 650°C for a standard AMS vaporizer. However, a qualitative observation of vaporizer surface temperature did not exhibit noticeable temperature gradient. Furthermore, the change in the $NO_x^+$ ratio due to temperature effect in Hu et al. (2017b) can not explain the variability observed during lens scan (a factor of 2). Therefore, the position-dependent decomposition on SV in Fig. 7 is likely caused mainly by the geometry effect. Although not investigated as thoroughly, a brand new standard vaporizer installed for the TI³GER campaign exhibited
the same feature as the vaporizer shown here (Fig. S5.3), indicating that position-dependent decomposition is likely a general feature of standard vaporizers.

As noted, the vaporizer center also exhibits a higher water signal (Fig. S5.4). The (background signal corrected) $H_2O^+$ to $NO_x^+$ signal ($\equiv NO^+ + NO_2^+$) ratio was as high as 0.15 at the center while the ratio ranged 0.05–0.08 on other parts of the vaporizer surface (Fig. S5.4). Note that unlike for typical AMSs, the background water signal in the CU AMS is negligible
due to the use of a cryopump, allowing for precise water signal measurements. Drewnick et al. (2015) reported $H_2O^+/NO_x^+$ ~ 0.23 from $NH_4NO_3$. The $H_2O^+$ signal is more likely from water formed from the thermal decomposition of $HNO_3(g)$ rather than particle water molecules remaining inside the dried calibration particle. If the majority of the water signal were coming from particle water, the $H_2O^+$ signal would be homogeneously distributed like the total nitrate signal. In contrast, when measuring nitrate in ET mode (single particle detection), $H_2O^+/NO_x^+$ was ~3% (~3 times lower than in MS mode; Fig. S5.4).
In ET mode, the time scale (210 µs) is not long enough to measure the full decay and enhanced background of the $H_2O^+$ signal. Since apparent inhomogeneities of the total nitrate signal were not observed (Fig. S5.2a–c), the effect of position-dependent nitrate decomposition on the calibration of AMS with $NH_4NO_3$ particles is limited. However, as discussed in Sect. 3.1.2.1, the yield of $NO_x^+$ signal from $NH_4NO_3$ decreases up to 3–4% for lower $NO_x^+$ ratio (higher thermal decomposition) for SV.

In Fig. 7b, the normalized $NO_x^+$ ratio is nominally measured outside of the vaporizer perimeter. This is because the
particle beam width is wider in the $PM_{2.5}$ lens and substantial aerosol signal (> 10 µg sm$^{-3}$ of nitrate) was detected even when the ADL was pointing outside of the vaporizer (cf. Fig. 10–11). Lens scans with the $PM_{2.5}$ lens do not show such an apparent center spot of low $NO_x^+$ ratio because the particle beam width is wider than for the $PM_1$ lens and HPL (see Sect. 3.2.3) smoothing the gradient of the $NO_x^+$ ratio. The disparity between the locations of the nominal vaporizer center (center position





of the projected vaporizer perimeter) and the actual vaporizer center (indicated by $NO_x^+$ ratio) asymmetricity of the $NO_x^+$ ratio image suggests that the vaporizer was tilted/misaligned (Fig. S5.8) during or after vaporizer installation.

When a CV is used and the particle beam is fully captured by the CV entrance, it is unlikely that the capture vaporizer exhibits position-dependent decomposition since the particles bounce inside the cavity for ~0.5 ms ensuring nearly full thermal equilibrium. However, when the particle beam hits near the edge where the temperature is supposedly colder, particles may bounce back and only partially vaporize with different fragments. At the vaporizer edge, the $NO_x^+$ ratio increases to 0.6–0.8

which is ~10 times higher than the center of CV and within the range of nominal $NO_x^+$ ratio with SV (Hu et al., 2017b; Xu et al., 2017).

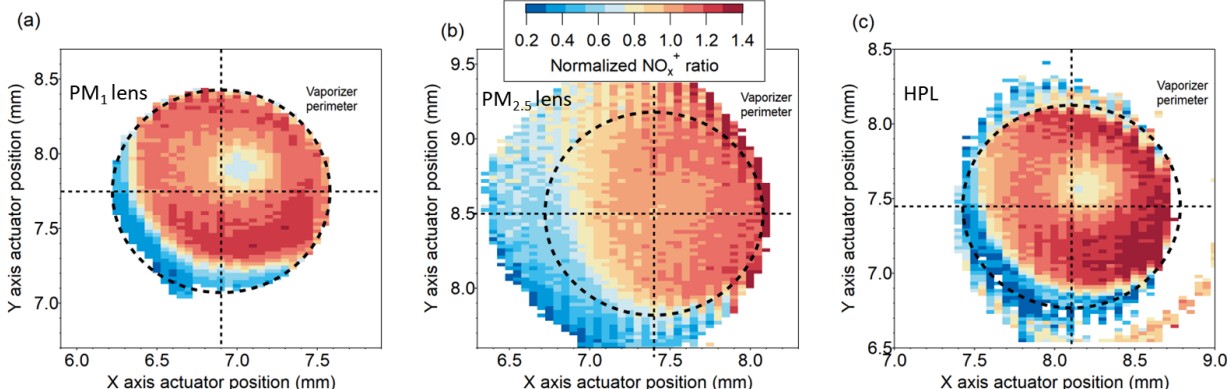

**Figure 7: Lens scan images of $NO_2^+/NO^+$ ratio ($NO_x^+$ ratio) normalized by the $NO_x^+$ ratio at the center of the vaporizer perimeter,**
**as a proxy for surface temperature obtained with monodisperse $NH_4NO_3$ particles from a (a) $PM_1$ lens (350 nm $d_m$), (b) $PM_{2.5}$ lens (350 nm $d_m$), and (c) HPL (800 nm $d_m$). The length scale is the same for all the plots. Only the data with nitrate concentration above 10 μg sm$^{-3}$ are displayed. For better visualization, the data are normalized by the ratio at the nominal center of the vaporizer.**

### 3.1.2.1. Implications of position-dependent decomposition for AMS organic nitrate quantification

The $NO_x^+$ ratio from other forms of nitrates, such as organic nitrates (pRONO$_2$) and NaNO$_3$, is lower than that of $NH_4NO_3$, and is often used to identify and quantify organic nitrate (Farmer et al., 2010; Fry et al., 2013; Day et al., 2022). In many previous chamber and field studies using the AMS with $PM_1$ lens, $NO_x^+$ ratios from $NH_4NO_3$ ranged from 0.2 to 1 (Day et al., 2022). In this study, the majority of observed $NO_x^+$ ratios during the lens scan with $PM_1$ lens ranged from 0.5 to 1 (when excluding vaporizer edge, Fig. S5.2d and Fig. S5.5). On the vaporizer edge (or slightly outside of the edge, left bottom side in

Fig. 7a), the low $NO_x^+$ ratio was lower (< 0.5). Only ~ 40% of the vaporizer edge was identified probably due to the misaligned vaporizer and blockage by the third skimmer (Fig. S5.8). This suggests that the position-dependent AMS response to nitrate fragmentation could partially account for the wide variability of the $NO_x^+$ ratio in previous studies. Day et al. (2022) presented the linearity between $NO_x^+$ ratios from pRONO$_2$ and $NH_4NO_3$ in previous studies with AMS. In that study, the variability of the $NO_x^+$ ratio from pRONO$_2$ and $NH_4NO_3$ among previous studies was tentatively attributed to vaporizer bias voltage drifts

or different MS tuning which can shift the $NO_x^+$ ratio of $NH_4NO_3$ by a factor of ~2. Day et al. (2022) proposed a ratio of ratio




(RoR) method that normalizes the $NO_x^+$ ratio from $pRONO_2$ by that of $NH_4NO_3$ in order to minimize instrumental variability that led to a more consistent $NO_x^+$ ratio among a variety of $pRONO_2$ species and mixtures.

We hypothesize that the literature variability in the $NO_x^+$ ratio also had a contribution from position-dependent decomposition. The linearity of $NO_x^+$ between $pRONO_2$ and $NH_4NO_3$ reported by Day et al. (2022) suggests that when the

particle beam impacts on or near the SV center (off-center), both $pRONO_2$ and $NH_4NO_3$ would be more (less) thermally decomposed by the SV. This hypothesis can be investigated by future work with lens scans using both monodisperse $pRONO_2$ and $NH_4NO_3$ particles and by observing the linearity of the position-dependent $NO_x^+$ ratio between the $pRONO_2$ and $NH_4NO_3$.

Apportionment of aerosol species based on ion ratios requires that calibrations are done routinely, and the species of interest interact with the vaporizer consistently. Changes in ion fragmentation ratios due to changes in the aerosol beam position

and/or width can be a potential source of errors in aerosol apportionment. This could happen if the ADL was moved to a different alignment after calibration or if the calibration and sample particle beam overlaps on the vaporizer changes due to the variability in particle beam width and/or beam position (*i.e.*, due to changes in particle sizes, see Fig. 11a). When using the $PM_{2.5}$ lens, due to its broader beam width compared to the $PM_1$ lens (Fig. 11), the $NO_x^+$ ratio is less sensitive to the position-dependent decomposition effect (Fig. 7b) and the uncertainty of $pRONO_2$ quantification due to this effect would be also less

significant. If aerosol apportionment is of interest and lens scanning is not available, our results suggest that it would be useful to periodically calibrate the AMS with multiple monodisperse test particle sizes to confirm a consistent response.

Takeuchi et al. (2024) reported that the nitrogen-containing moiety mass concentrations in $pRONO_2$ from AMS and CPC were best matched when CPC-based mass was calculated based on the $–NO_2$ group not $–ONO_2$ group. This observation ultimately translates to ~35% lower AMS nitrate sensitivity to $pRONO_2$ (*i.e.,* ~ 35% lower $NO_x^+$ signal per nitrate mass) than

for equivalent $NH_4NO_3$. They discussed that the thermal decomposition of $pRONO_2$ on the vaporizer yields mostly $NO_2(g)$ (hence lower $NO_x^+$ ratio, 0.1–0.3 in that study), while $NH_4NO_3$ yields more $HNO_3(g)$ (higher $NO_x^+$ ratio, 0.6 in that study). They attributed the cause of lower nitrate sensitivity to the nitrogen moiety being $NO_2$ (not $NO_3$) after thermal decomposition and suggested that the measured nitrate from $pRONO_2$ needs to be scaled up by ~35% to properly represent the mass of the -$ONO_2$ group. After the thermal decomposition, $NO_2$ might yield lower $NO_x^+$ signal than for $HNO_3$ since ionization efficiency

of EI tends to be proportional to molecular weight (Jimenez et al., 2003).

We evaluated the relative sensitivity of nitrate using the $NO_x^+$ ratio as a proxy for the degree of thermal decomposition and the resulting distribution of nitrogen oxides. In Fig. S5.5, nitrate mass normalized by ammonium was plotted against the $NO_x^+$ ratio from a lens scan of a SV and a manual lens scan of a CV (Hu et al., 2017b) with $NH_4NO_3$. Ammonium does not show signs of thermal fragmentation in the AMS (indicated by the stability of the ion ratios of the $NH_x$ family), while the $NO_x^+$

ratio varied by an order of magnitude. Hence, the metric (nitrate normalized by ammonium) was used to track the change of nitrate sensitivity due to thermal decomposition. The CV entrance edge and off-center positions of the SV showed a similar $NO_x^+$ ratio (~ 0.65–1). The SV center ($NO_x^+$ ratio ~ 0.5-0.65) and CV inner cavity ($NO_x^+$ ratio ~ 0.05–0.1) showed ~3% and ~10% lower nitrate sensitivity, respectively, compared to SV off-center positions. Thus, increasing thermal decomposition may lead to slightly lower sensitivity of nitrate. This suggests that, if $pRONO_2$ predominantly yields $NO_2(g)$ after thermal



decomposition on SV (yielding $NO_x^+$ ratio ~0.1), the sensitivity of nitrate from $pRONO_2$ would be 7–10% lower than $NH_4NO_3$. However, that 7–10% estimated lower sensitivity is not consistent with the ~35% lower nitrate sensitivity of $pRONO_2$ compared to $NH_4NO_3$ reported by Takeuchi et al. (2024). This discrepancy suggests that there may be factors, other than $NO_2(g)$ being the primary source for $NO_x^+$ ions, that caused the lower nitrate sensitivity from $pRONO_2$ observed by Takeuchi et al. (2024). Thus, the AMS nitrate sensitivity to $pRONO_2$ requires further evaluation by additional laboratory studies and

intercomparisons with instruments with fundamentally different working principles (Fry et al., 2013; Kenagy et al., 2021; Day et al., 2022).

### 3.1.2.2. The implication of position-dependent decomposition on OA characterization

During a lens scan using oleic acid particles, higher $f_{H2O+}$ and $f_{CO2+}$ (fraction of $H_2O^+$ and $CO_2^+$ ion among the organic aerosol

ions) values were observed at the vaporizer center (Fig. S5.6). $f_{CO2+}$ ($f_{44}$ in unit mass resolution) is used as an indicator of OA oxidation or age (Ng et al., 2011) and used for parameterization for the atomic O/C ratio of ambient organic aerosol (Aiken et al., 2008; Canagaratna et al., 2015). Canagaratna et al. (2015) showed that $f_{CO2+}$, $f_{CO+}$, and $f_{H2O+}$ of OA standards depended on SV temperature, with their response varying among the tested compounds.

Given the observed higher thermal decomposition of $NH_4NO_3$ at the center of the vaporizer, the increase in $f_{CO2+}$

could be attributed analogously to more efficient thermal decarboxylation of oleic acid. At the center, $f_{H2O+}$ went up to 0.09. During the lens scan, $f_{CO2+}$ and the O/C ratio (parameterized from $f_{CO2+}$) varied by up to a factor of ~4 and a factor of ~2, respectively, reaching a maximum at the center of the vaporizer. On the other hand, $m/z$ 55 fraction ($f_{55}$, mostly $C_4H_7^+$), which is often used as an indicator of hydrocarbon-like OA, was not noticeably affected by the location of particle impact (Fig. S5.6e).

Ambient OA may be subject to position-dependent decomposition as well. For example, $f_{44}$ was systematically higher for CV compared to SV AMSs for ambient OA from various field campaigns (Hu et al., 2018) indicating the enhanced thermal decomposition of oxidized OA increased $f_{44}$ further. Since the enhanced thermal decomposition at the SV center has a similar effect of increased $f_{44}$ in OA measurements, this effect might have contributed to the observed variability of $f_{44}$ across multiple SV-equipped ACSM/AMSs when sampling the same ambient aerosol (Crenn et al., 2015; Fröhlich et al., 2015). More studies

are needed to better understand the implications of position-dependent decomposition on the characterization of ambient OA.

### 3.2. Particle beam profiling using the 2D-SR-BWP method

### 3.2.1. Analysis of 2D-SR-BWP profiles

The 2D-SR-BWP analysis provides a novel and fast, quantitative measurement of the particle beam profile, *i.e.,* the particle beam width and center position along two orthogonal axes ("X" and "Y", corresponding to a rotation of the BWP stage of 0

deg and 90 deg, respectively, see Sect. S6). Fig. 8 shows results from 2D-SR-BWP measurements for a $PM_{2.5}$ lens coupled to a PCI (PCI-C) using polydisperse $NH_4NO_3$ using $m/z$ 17 ion ($NH_3^+$). Fig. 8a-b shows the time series of the particle size





distribution during the BWP wire scan for the X and Y BWP axes. At each internal wire position, the beam may be partially or completely blocked and then the wire moves away from the beam path to measure the reference distribution of the aerosol source. The resulting particle signal attenuation factor ($A$, see Eq. 6) as a function of $d_{va}$ at each BWP wire position wire is shown in Fig. 8c-d as well as particle mass size distribution generated from the nebulizer measured by AMS PToF mode. Above ~1.5 μm $d_{va}$, signal is limited by inlet transmission substantially decreasing signal to noise. $A$ as a function of BWP position is at each $d_{va}$ shown in Fig. 8e-f.

Beam position differences up to ~0.2 mm (~0.27 mm on the vaporizer plane) were observed previously between two different sizes (110 nm vs 320 nm $d_{va}$) for $(NH_4)_2SO_4$, $NH_4NO_3$, and oleic acid particles when using a $PM_1$ lens (Huffman et al., 2005). More recently, the beam position shift vs. particle size has also been observed with a custom-designed ADL although not quantified (Clemen et al., 2020). In Fig. 8e-f, the beam center positions varied significantly (up to ~1 mm) depending on the particle diameter over the x-axis while along the Y axis the beam center barely changed. This indicates that BWP measurement along only one axis could fail to capture the variability in particle beam position and width. The particle beam along the X axis was separated into two beams having double peaks ~160 nm $d_{va}$ (see the thick red line in Fig. 8e). The measured beam widths were broader at smaller diameters on the X axis while the widths on the Y axis varied less compared to the X axis over the measurable size range. In Sect. 3.2.3, we show that variations in beam center position and beam width vs. particle size are observed for all ADLs tested.





**Figure 8: Top: Time Series of aerosol mass distribution at m/z 17 (NH₃⁺, a major ammonium fragment ion) for a 2D-SR-BWP measurement with a PM₂.₅ lens using polydisperse NH₄NO₃ particles. Middle: Normalized signal attenuation factor (A) at each BWP position as a function of $d_{va}$, while scanning along the x (left) and y (right) BWP axes. The grey dotted line represents the reference NH₄NO₃ size distribution (when the BWP wire is not blocking the beam) measured by PToF mode. Bottom: Normalized signal attenuation for each particle diameter bin along the x (left) and y (right) BWP axes as a function of BWP wire position. (e-f) illustrates that the particle beam width and center position can vary depending on particle size traveling through an ADL. The BWP position is the BWP wire position whose zero position is aligned with the vaporizer center (Sect. S6). For diameters where the input concentrations are higher, the signal-to-noise of the measured attenuation factor is higher. The stability of the aerosol source on a per-size-bin basis was within 10% (1 σ) below 1300 nm $d_{va}$ (Fig. S7.2).**





### 3.2.2. Comparison of the beam width obtained from regular BWP and SR-BWP

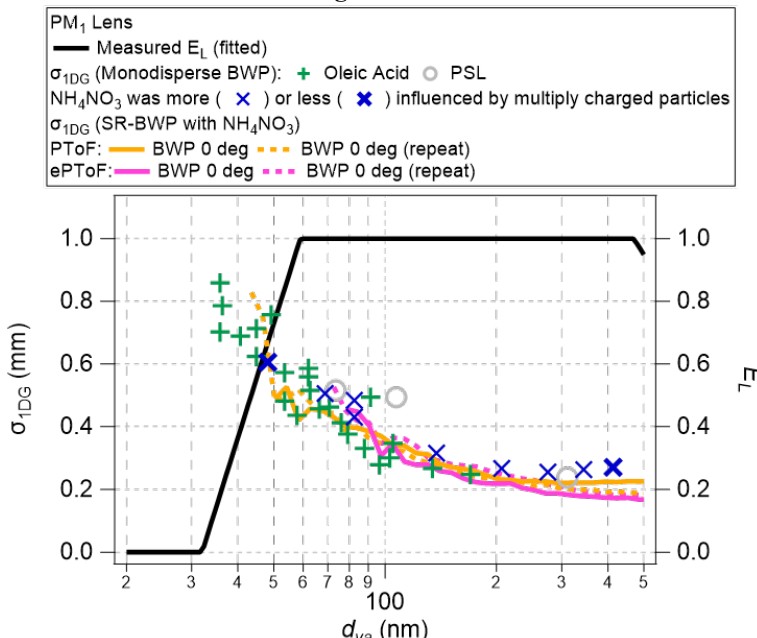


**Figure 9: Particle beam width measurements with a PM$_1$ lens (left axis) by BWP using monodisperse PSLs, oleic acid, and NH$_4$NO$_3$ particles. Also shown is the SR-BWP data obtained from polydisperse NH$_4$NO$_3$ particles, confirming the good agreement between the two techniques. The $E_L$ for this PM$_1$ lens (per Fig. 14) is shown on the right axis. The BWP measurement with monodisperse particles was done with a PM$_1$ lens coupled with PCI-C and SR-BWPs were conducted with the same PM$_1$ lens but without PCI.**
**Monodisperse particle beam width measurements of NH$_4$NO$_3$ particles are influenced by multiply charged particles (Fig. S1.3). The decrease in $E_L$ when the measured beam width ($\sigma_{1DG}$) increases above 0.5 is consistent with the model prediction (Fig. 4c). The agreement between the SR-BWP NH$_4$NO$_3$ and monodisperse oleic acid BWP results suggests that the particle transmission below $d_{va}$ < 100 nm can be inferred by performing SR-BWP measurements in the laboratory or in the field.**

Fig. 9 shows particle beam widths for the PM$_1$ lens measured in two ways, as a size-resolved BWP (SR-BWP) with polydisperse particles in both PToF and ePToF modes and as a regular BWP measurement with monodisperse particles. Both BWP and SR-BWP were conducted along only one axis. Oleic acid, NH$_4$NO$_3$, and PSLs were used for monodisperse BWP measurements and polydisperse NH$_4$NO$_3$ was used for SR-BWP measurements. Oleic acid particles below 120 nm $d_{va}$ were generated from the evaporation-condensation system and DMA (Sect. S1) and allowed us to characterize particle beam width

all the way down to the ADL's transmission limit. The monodisperse NH$_4$NO$_3$ particles included some multiply charged particles. The presence of multiply charged particles (larger diameter and narrower beam width) will reduce the apparent beam width while the different beam positions for different sizes (Fig. 11.b) may increase the measured width. In Fig. 9, bold X's indicate BWP measurements that were minimally influenced by multiply-charged particles (multiply-charged mass was less than 15% of singly-charged mass) while light X's indicate measurements with more multiply-charged particles. There is good

agreement between the measurements with more or less multiply-charged contribution. In addition, the width from DMA-generated NH$_4$NO$_3$ showed a reasonable agreement with oleic acid particles.



The beam widths measured from monodisperse BWP and $NH_4NO_3$ from SR-BWP agreed well. Thus, the use of PToF mode for BWP measurements, which involves a very high concentration of aerosol input (typically several mg sm$^{-3}$) vs. BWP with monodisperse input (a few to tens of µg sm$^{-3}$), does not appear to bias the results. Also, the SR-BWP results with PToF and ePToF were nearly identical, indicating that the different size information reduction procedures (Fig. 9) do not affect the results. The SR-BWP results shown in the later sections are performed in PToF mode (rather than ePToF) due to the lower retrieval noise (and hence higher per-bin stability). SR-BWP analysis provides the same beam width information as obtained from the conventional monodisperse BWP operation but has significant benefit in that it provides size resolved information, in particular for small sizes that are difficult to access with monodisperse measurements.

Fig. 9 shows the fit to the measured $E_L$ for the PM$_1$ lens (Fig. 14a). The measured $E_L$ decreases as beam width increases ($\sigma_{1DG}$ above ~0.5 mm) at lower $d_{va}$. $E_L$ is 50% around 45 nm $d_{va}$ where $\sigma_{1DG}$ is 0.6-0.8 mm (0.75-0.8 mm with SR-BWP). These are consistent with beam modeling results (Fig. 4c) where the estimated transmission efficiency starts decreasing above ~0.5 mm $\sigma_{1DG}$ and is 50% at ~0.77 mm $\sigma_{1DG}$, assuming the beam center position is close to the vaporizer center. These results show that the $E_L$ of the PM$_1$ lens below 100 nm $d_{va}$ is mainly driven by the beam widening of small particles at the lens nozzle.

### 3.2.3. Particle beam profiles for different ADLs

Fig. 10 shows 2D-SR-BWP attenuation vs. BWP wire position and $d_{va}$. Particle beam profiles from the PM$_1$ lens, PM$_{2.5}$ lens, and HPL were calculated using 2D-SR-BWP measurements, at $P_{Lens}$ of 2 mbar, 5.1 mbar, and 21 mbar, respectively. Here, beam profile refers to a gaussian fit to the attenuation data at each $d_{va}$ bin (*e.g.,* Fig.8e-f). See Fig. S10.1 for the signal attenuation prior to Gaussian fitting. The lenses were tested with a single critical orifice ($d_{CO,std}$ = 120 µm), without pressure controlled inlets. The 2D-SR-BWP measurements were conducted along the X and Y BWP axes for the PM$_{2.5}$ lens and HPL, while the measurements from the PM$_1$ lens are available only for the X axis and an axis at 30 degrees from the X axis. Fig. 11 summarizes the measured beam widths ($\sigma_{1DG}$) and the relative beam center positions.

The PM$_1$ lens data show the tendency of beam broadening at smaller particle sizes due to Brownian motion at the nozzle expansion while showing better beam collimation at larger sizes (Fig. 11.a). The beam center position shifted as a function of $d_{va}$, up to ~0.3 mm (Fig. 11.b) which is equivalent to ~0.4 mm shift at the vaporizer (11% of vaporizer diameter). This is not negligible, considering the transmission efficiency (Fig. 4c) and position-dependent fragmentation on the vaporizer (Fig. 7a). However, the shift in beam center position for the PM$_1$ lens was smaller than for the HPL and the PM$_{2.5}$ lenses discussed next.

The results for the PM$_{2.5}$ lenses show large variation both in beam width and center position as a function of $d_{va}$. The particle beam position from the PM$_{2.5}$ lens shifted up to ~0.65 mm (~0.9 mm in the vaporizer plane and hence ~25% of the vaporizer diameter), similar to the HPL. Interestingly, unlike the PM$_1$ lens and HPL, the PM$_{2.5}$ lens exhibited an increased beam width above ~400 nm $d_{va}$ (also see Fig. S10.3c and Fig. S16.1c–d). This indicates that optimizing the lens alignment is important not only for maximizing the small particle transmission but also for midrange sizes (300–700 nm $d_{va}$) as well. The



different beam widths from the PM$_{2.5}$ lens along the X and Y BWP axes indicate that the cross-section of these particle beams is elliptical (especially for $d_{va} < 80$ nm and $d_{va} > 400$ nm). Note that the data quality of 2D-SR-BWP results for the PM$_{2.5}$ lens in a single critical orifice setup was compromised compared to the results from the PM$_1$ lens and HPL in terms of particle size range, due to issues operating the aerosol source at large sizes during those experiments (Fig. S10.1c–d). The reproducibility of the BWP wire control during the Y axis measurements adds additional uncertainty to this measurement (Fig. S10.1d).

However, the comparison with repeated measurements (Fig. S10.2–3) suggests that the beam broadening in the 300–700 nm $d_{va}$ range and the large shift in beam center position are the features of the PM$_{2.5}$ lens. These features were also observed from a different Aerodyne's PM$_{2.5}$ lens with improved nozzle design with ruby orifice (data not shown).

The HPL again shows a monotonic decrease in beam width for larger particles, as for the PM$_1$ lens. Below 100 nm, the particle beam from the HPL was broader than for the PM$_1$ lens, which is consistent with the lower $E_L$ of the HPL in that

particle size range (Fig. 14c). The particle beam position from the HPL shifted noticeably depending on the particle size, up to ~0.65 mm, which is equivalent to ~0.9 mm at the vaporizer (~25% of the vaporizer diameter). Thus, for small particle transmission, the center shift would be as important as the beam broadening. Maximizing particle transmission for the broad range of particle sizes accessible with the HPL will benefit from both information on the beam profile in two dimensions and an accurate lens alignment tool.

The shifts in beam focusing and pointing may depend on the ADL design, operating pressure, and upstream plumbing (*e.g.*, use of PCI vs. single critical orifice), and other factors. The beam widths of the PM$_{2.5}$ lens coupled with PCI-D were very similar to those without a PCI, indicating the use of PCI-D does not significantly influence the beam focusing of the ADL (Fig. S16.1c). Direct comparison of the relative beam position with the bare PM$_{2.5}$ lens is complex since the bare lens measurements were conducted before the TI$^3$GER campaign and the rotational orientation of the PM$_{2.5}$ lens is likely different which means

the axes in Fig. S16.1d with and without PCI are not the same. Even for the same type of ADL, imperfections in the machining process could lead to such irregular beam focusing/pointing, although some features may remain qualitatively the same, such as beam broadening of ~500 nm $d_{va}$ particles from PM$_{2.5}$ lens.

In BWP measurements, the measured $\sigma_{1DG}$ cannot be narrower than a certain width ($\sigma_{1DG,min} \sim 0.18$ mm, Fig. 3b) because the width of the wire itself limits the minimum width that can be measured (Sect. 2.3.3). In other words, a beam width

narrower than $\sigma_{1DG,min}$ will appear as $\sigma_{1DG,min}$ in the BWP measurements (Fig. 11a). Although the actual beam width can be estimated through modeling as was done in this work (Fig. 4b) and Huffman et al. (2005), the modeled estimation becomes more uncertain for $\sigma_{1DG} < 0.25$ mm. On the other hand, beam width measurement based on lens scans can be more sensitive to narrower beam widths providing complementary measurements to SR-BWP (Sect. 3.1.1 and Fig. S5.7). The lens scan based beam width measurement shows that the beam width from the PM$_1$ lens can be as narrow as 0.035 mm (at ~480 nm $d_{va}$) which

is about an order of magnitude narrower than the beam width from the PM$_{2.5}$ lens. The relative center positions of monodisperse particles from lens scans (Fig. 11b) were also measured from lens scans with monodisperse particles of different $d_{va}$ (Fig S5.6c). The beam position shift vs. $d_{va}$ observed from lens scan and 2D-SR–BWP were similar (Fig. 11b).



Clemen et al. (2020) measured particle beam width for multiple PSL sizes by mechanically tilting the lens, conceptually similar to the lens scanning in this study. They characterized a custom-designed ADL consisting of conical
orifices for particle collimation. The aerosol beam width measurement by two perpendicularly aligned detection lasers (DL1 and DL2) provided beam width information along the two perpendicular axes. The beam width comparison between the PM$_1$ lens and that custom ADL shows the beam width of the PM$_1$ lens to be similar to or narrower than the width of the custom ADL in the 200–500 nm $d_{va}$ range (Fig. 11a). The beam width measurement from a PM$_{2.5}$ lens in Clemen et al. (2020) (Fig. 16 in that paper) increased at $d_{va} > 300$ nm which is qualitatively consistent with our results. Beam width from HPL was similar
to that from the custom ADL in 400–1200 nm $d_{va}$ range.



**Figure 10: 2D-SR-BWP signal attenuation measured with polydisperse NH₄NO₃ using a PM₁ lens (a,b), PM₂.₅ lens (c,d), and HPL (e,f) in X (left column, without BWP stage rotation) and Y (right column, BWP stage 90 deg rotated, except for PM₁ lens) axis. The 1st row of the right column is only at a 30 degree rotation from the X axis of the PM₁ lens. Here, measured signal attenuations were fitted with a Gaussian (see Fig. S10.1 for results without Gaussian fit). The dotted lines represent the vaporizer edges projected to the BWP plane.**






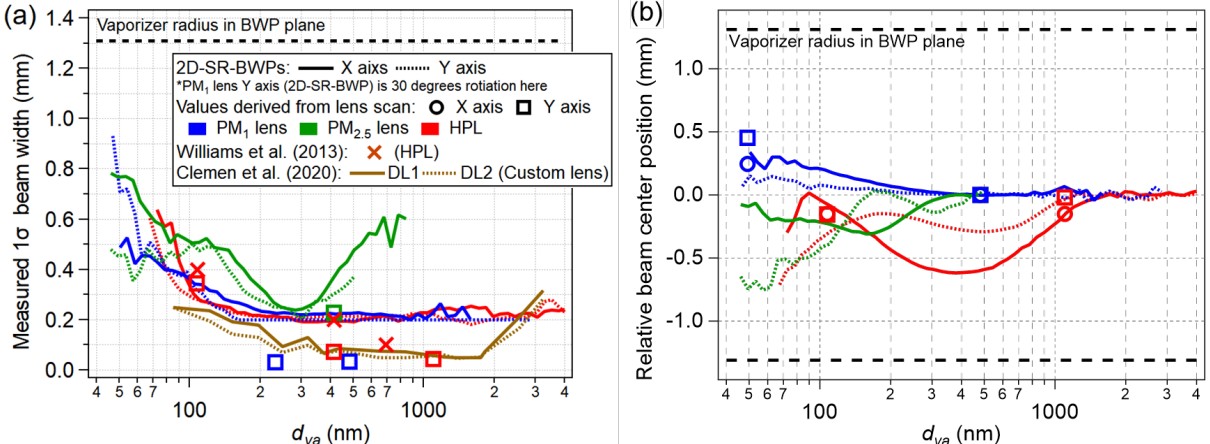

**Figure 11: Compilation of (a) beam width ($\sigma_{1DG}$) and (b) relative beam center position (relative to largest measured particles) as a function of $d_{va}$ for the PM$_1$, PM$_{2.5}$, and HPL lenses, measured in X and Y axis at the BWP plane. Beam width measurements using a custom-designed ADL (Clemen et al. 2020) and PSL particles are shown for comparison width ADLs from Aerodyne. Clemen et al. (2020) measured beam width by mechanically tilting lens and the aerosol beam in two perpendicularly aligned detection lasers (DL1 and DL2). The widths from Clemen-ADL were scaled to reproduce the width after traveling the same distance as the ADLs in this work. In (b), the relative beam positions measured using lens scan of larger particles were matched to the relative position from 2D-SR-BWP of the same $d_{va}$.**

## 3.3. Validation of particle beam model with TE and particle deposition

In this section, we investigate a case study of an HPL equipped with PCI-D ($d_{CO,up}/d_{CO,down}$ = 500/400 µm, $P_{Lens}$ = 21 mbar) and compare the TE estimated by the particle beam model constrained by 2D-SR-BWP (Sect. 2.3.3) with the measured TE. Fig. 12a shows the beam center position vs. $d_{va}$. TE in the model accounts for particle losses due to failing to hit the vaporizer due to wide beam widths (especially for smaller particles) and/or the center position of the beam being too far off from the vaporizer center. The modeled TE here only describes transmission losses after the exit of the lens and hence does not account for the loss of particles by impaction in either the PCI or within the lens itself, which affects the transmission of particles larger than ~500 nm $d_{va}$ (Fig. 12b).

Fig. 12b also shows the measured particle beam center positions and beam width ($\sigma_{1DG}$) as measured with 2D-SR-BWP. The model then simulates the BWP measurement of the Gaussian beam to retrieve the experimental particle beam parameters. The slight mismatch between the input position/width and the simulated position/width for the smallest particles ($d_{va}$ < 150 nm) indicates that the measured beam is too broad or too close to the vaporizer edge and that the reconstructed beam is biased toward more centered positions with a narrower width. The mismatch also indicates that, during the 2D-SR-BWP measurements, the measured position and width may have been potentially biased in the same direction for the same reasons. Thus, when the measured and simulated positions/width mismatch, the simulated TE can be overestimated. Although the higher modeled TE below 150 nm $d_{va}$ may be attributed to the measurement bias of beam width, the modeled TE follows the decreasing trend of measured TE providing a reasonable estimation of the lower-end transmission curve.





TE modeling can be useful to estimate the small particle transmission, which is important when measuring Aitken mode particles, without having to generate monodisperse aerosols below 200 nm $d_m$ (without doubly charged particles) which

is challenging outside of the laboratory. Another useful model feature is that it can diagnose particle losses due to failing to hit the vaporizer when the beam center positions approach the edge of the diameter (see Sect. 3.5.1).

For further validation, the particle beam model was applied to reconstruct particle deposition images taken by Aerodyne as part of the lens quality control process. Particle deposition pattern images in Fig. 13a-c were made by impacting polydisperse $NH_4NO_3$ aerosol sampled through the $PM_{2.5}$ lens and HPL, 41 cm after the exit of the lens on a glass surface

(Sect. S8). The white circle indicates the scaled perimeter of the standard vaporizer. The HPL deposition image shows the shift of center position between small particles (wider width) and larger particles (smaller width) which is consistent with Fig 11b.

Simulated particle deposition images for each lens for each lens are shown in Fig. 13c-d. For these model runs, the size distribution from the TSI atomizer measured in our laboratory (Fig. S7.1a) was combined with the beam profiles shown in Fig. 11, but the profiles were adjusted to the center of the vaporizer. In the simulation, the total deposition was normalized

to the highest intensity and the contour lines indicate 20, 40, 60 and 80% of the maximum particle deposition. The particle deposition simulations are generally consistent with the deposition images. Some differences between the simulations and measurements may result from the potential differences in the aerosol size distributions used in both tests. Also, while the model assumes the deposited aerosol mass is proportional to the intensity in the image, the particle image intensity may not be proportional to the deposited mass.


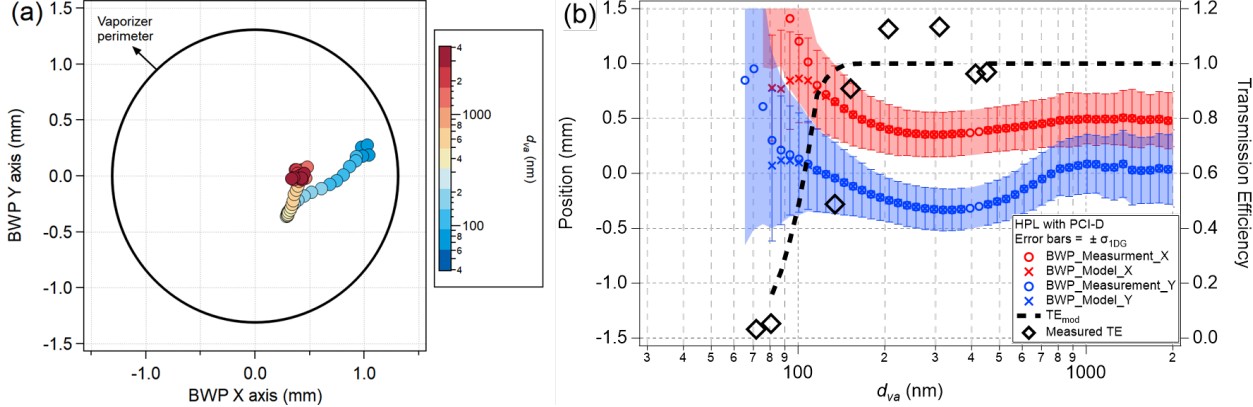

**Figure 12: (a) Particle beam center position trajectories from HPL with PCI-D as a function of $d_{va}$, as measured by 2D-SR-BWP. The nitrate concentration during the lens scan is also shown as an image plot in the background. The vaporizer edge is shown as a dotted green line, as determined from the lens scan. (b) Modeled and measured TE of the inlet. Also shown are measured center**
**positions and reconstructed beam center positions in the model. Bars and shading represent 1 σ of the Gaussian fitting.**

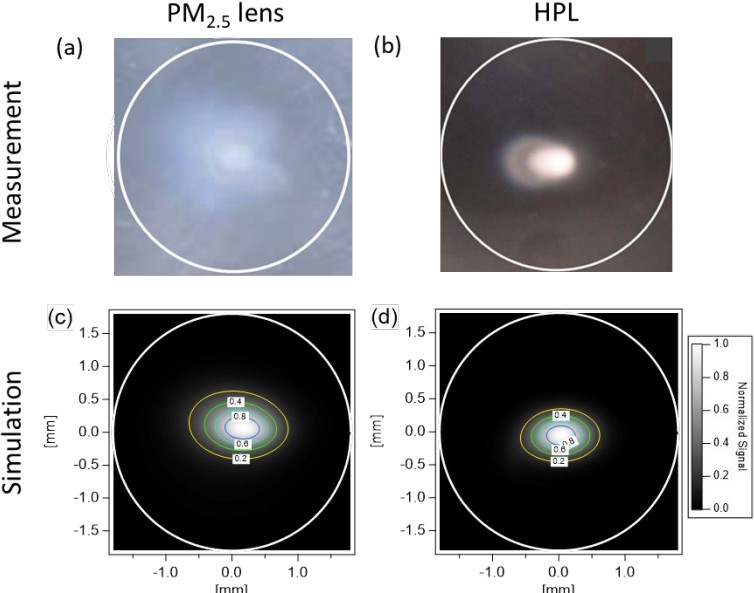

**Figure 13: Deposition patterns of polydisperse NH₄NO₃ transmitted by the lens system for (left) PM₂₅ lens, (right) HPL with spot check images (top) and simulated particle deposition (bottom). The contour lines indicate where the aerosol deposition is 0.2, 0.4,**
**0.6, and 0.8 fraction of the highest aerosol deposition. The HPL spot check image was adopted from Williams et al. (2013).**

### 3.4. Laboratory tests for characterization of the aircraft inlet

#### 3.4.1. $E_L$ of standalone PM₁, PM₂.₅, and HPL lenses

$E_L$ of three standalone ADLs (PM₁, PM₂.₅, and HPL) at 2, 5.1, and 21 mbar $P_{Lens}$ respectively, were measured with monodisperse particles of different compositions (Sect 2.1 and 2.4). Typical atmospheric pressure in Boulder was ~820 mbar
and the pressure in the AMS aerosol sampling line was ~790 mbar. A single CO ($d_{CO,std}$ = 120 μm) was used coupled with EV-C to minimize particle losses after the CO$_{std}$ (Fig. S3.2). $E_L$ was calculated by comparing the aerosol mass measured by AMS and CPC (Sect. 2.4). The particle speed calibration curve was fitted following Eq. 4. In this work, measured $E_L$ linearly increases in log($d_{va}$) space as $d_{va}$ increases (usually below 200 nm $d_{va}$), and plateaus afterward around 1, and then linearly decreases. Hereafter, $d_{va}$ where TE is 50% are referred to as $d_{va,50,low}$ ($d_{va,50,high}$) on the lower (higher) side of the TE curve.

The measured $d_{va,50,low}$ for the PM₁ lens in this work ($d_{va,50,low}$ ~ 47 nm) was lower than the reported value by Knote et al. (2011) ($d_{va,50,low}$ ~ 63 nm) which is an averaged value from multiple studies (DeCarlo et al., 2004; Park et al., 2004; Cross et al., 2007; Vaden et al., 2011). Zhang et al. (2004a) reported $d_{va,50,low}$ ~ 45 nm, which is consistent with this work, using data from a very strong new particle growth event. The measured $d_{va,50,low}$ of the PM₁ lens in this work was noticeably lower than the value reported by Liu et al. (2007) ($d_{va,50,low}$ ~ 63 nm and 95 nm for 780 mbar and 1013 mbar ambient pressures,
respectively) (Fig. 14a). It is unclear what causes the observed differences, with possible explanations including differences between different actual lenses, larger deviations for beam pointing of small particles for some lenses, and/or measurement inaccuracies. The measured $d_{va,50,high}$ ~ 830 nm in this work was lower than the recommended value in Knote et al. (2011)



($d_{va,50,high} \sim 1060$ nm). It was higher than $d_{va,50,high}$ measured at 1013 mbar (650 nm) and similar to $d_{va,50,high}$ measured at 780 mbar (900 nm) reported in Liu et al. (2007). On the other hand, it is close to the lower $E_L$ PCI setup used during SEAC⁴RS reported in Hu et al. (2017b) and Guo et al. (2021).

The PM$_{2.5}$ lens shows slightly worse transmission for small particles than the PM$_1$ lens (Fig. 14b), which is consistent with the relative beam widths determined for small particles (Fig. 11a). Xu et al. (2017) reported $d_{va,50,low} \sim 150$ nm, significantly worse than the measurements in this study ($d_{va,50,low} \sim 55$ nm). Although the exact reasons behind the difference are unclear, it is possible that the lens alignment in Xu et al. (2017) might not have been in an optimal position for small particle sampling. The $E_L$ measurements in Xu et al. (2017) were performed with an Aerosol Chemical Speciation Monitor with a capture vaporizer, whose particle entrance diameter is smaller (2.54 mm) than for the standard vaporizer (3.8 mm). Given the use of a vaporizer with a smaller acceptance angle and the fact that the PM$_{2.5}$ lens can have more variation in position vs. $d_{va}$ compared to the PM$_1$ lens (Fig. 11b), the particle beams for smaller diameters might not have been well captured by the capture vaporizer in Xu et al. (2017). The PM$_{2.5}$ lens shows much better transmission for large particles ($d_{va,50,high} \sim 2.5$ µm) than the PM$_1$ lens, consistent with previous measurements (Xu et al., 2017; Molleker et al., 2020). In certain urban environments, aerosol mass measurements with a PM$_{2.5}$ lens were up to ~30% higher than measurements with a PM$_1$ lens due to the broader particle transmission size range (Elser et al., 2016; Joo et al., 2021; Li et al., 2023; Liu et al., 2024).

The HPL operates at the highest $P_{lens}$ (21 mbar) followed by the PM$_{2.5}$ lens (5.1 mbar) and PM$_1$ lens (2 mbar). The measured particle speeds in vacuum were generally higher for lenses with higher $P_{Lens}$, with particles exiting the HPL being the fastest (Fig. 14d). The $P_{Lens}$ in this work was ~14% higher than the $P_{Lens}$ in Williams et al. (2013) (Table S9.1) for the same lens flow. The $E_L$ of small particles ($d_{va,50,high} \sim 120$ nm, Fig. 14c) was worse than for Williams et al. (2013), which is consistent with the laboratory tests that showed larger $d_{va,50,low}$ when $P_{Lens}$ is higher (Fig. S13.2). The high transmission for large particles ($d_{va,50,high} > 1500$ nm) was consistent with Williams et al. (2013).

Williams et al. (2013) modeled the difference in $E_L$ between the HPL with and without an EV (or relaxation chamber) which significantly enhanced the transmission above 1 µm $d_{va}$. In this study, while the particle transmission of the EV-C (a different volume than in Williams et al., 2013; Fig. 5a) was not experimentally characterized, $E_L$ from the HPL coupled with EV-C appeared to be similar (up to ~1.5 µm $d_{va}$) to the $E_L$ result from Williams et al. (2013). Additionally, fluid dynamic modeling of particle transmission of the EV-C with 120 µm CO$_{std}$ at 5, 13, 17 mbar $P_{Lens}$ suggests that $d_{va,50,high}$ for this expansion volume is > 4500 nm (Fig. S3.2) indicating the measured $E_L$ of the HPL and the PM$_{2.5}$ lens reported here was likely not limited by the transmission of EV-C. For the PM$_1$ lens, the $d_{va,50,high}$ reported here was smaller by ~150 nm than the $d_{va,50,high}$ of the PM$_1$ lens equipped with PCI-C as reported by Guo et al (2021) and confirmed by this work (Fig. 15a,d). This suggests that the transmission of EV-C operated at ~2 mbar (with 120 µm CO$_{std}$) potentially limits the large particle transmission of the PM$_1$ lens. Additionally, particle losses inside plumbing upstream of the CO$_{std}$ could affect the $d_{va,50,high}$ (Liu et al., 2007). In general, the large particle transmission is affected by the geometry of the plumbing setup upstream of ADL and thus, the differences in the measurements of $d_{va,50,high}$ (Fig. 14a-b) can be partially attributed to the plumbing geometry upstream of ADL.

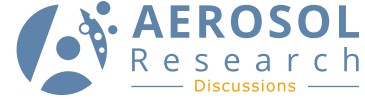

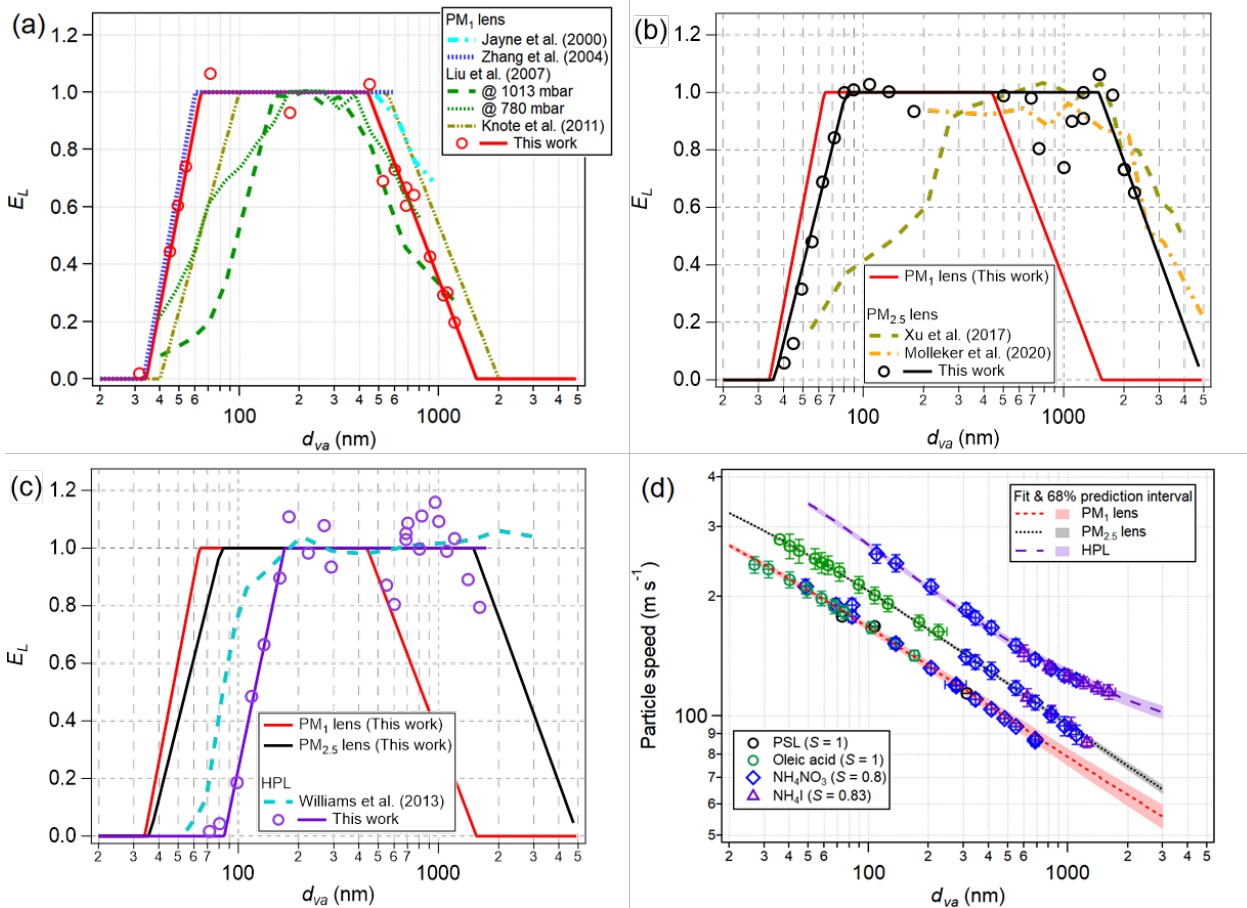

**Figure 14:** Measured and literature $E_L$ of standalone (a) PM$_1$ lens, (b) PM$_{2.5}$ lens, and (c) HPL. (a) PM$_1$ lens $E_L$ from Zhang et al. (2004a) was estimated from field measurements. $E_L$ shown by Liu et al. (2007) are their measurements at 780 mbar and 1013 mbar ambient pressure. $E_L$ from Knote et al. (2011) is a recommendation based on the average $E_L$ used in several studies (DeCarlo et al., 2004; Park et al., 2004; Cross et al., 2007; Vaden et al., 2011). (b) PM$_{2.5}$ lens $E_L$ from Xu et al. (2017) was measured by an Aerodyne Aerosol Chemical Speciation Monitor with a capture vaporizer. Molleker et al. (2020) used the MPIC Aircraft-based Laser ABlation Aerosol MAss spectrometer (ALABAMA) and CPI (see Sect. 3.4.2). (c) The HPL used in Williams et al. (2013) and in this study are the same physical lens. More details can be found in Sect. S9. (d) Particle speed vs. $d_{va}$ during ePToF calibration of the three lenses with oleic acid, PSL, NH$_4$NO$_3$, and NH$_4$I. For details of S of NH$_4$I, see Sect. S4. The measured particle speeds were faster for lenses with higher operating pressure, for the particles of the same $d_{va}$.

### 3.4.2. Transmission efficiencies of ADL coupled with PCI

Figure 15a–c shows the measured TE vs $d_{va}$ with PCI-C + PM$_1$ lens, PCI-C + PM$_{2.5}$ lens, and PCI-D + PM$_{2.5}$ lens inlet setups, respectively. Hereafter, these combinations of PCI and ADL will be referred to as PCI-ADL, for example, PCI-C-PM$_1$, PCI-C-PM$_{2.5}$, and PCI-D-PM$_{2.5}$. These inlet setups were tested with various choices for the bottom critical orifice, 220, 300, 350, and 400 μm $d_{CO,down}$, resulting in 250, 125, 96, and 72 mbar for $P_{PCI}$, respectively, to maintain the nominal $P_{lens}$ (2 mbar for PM$_1$ lens and 5.1 mbar for PM$_{2.5}$ lens). The use of a PCI did not noticeably affect low-end particle transmission, indicating



that the residence time was short enough to prevent diffusional losses. Maximum TE plateaus at 1 within the precision of the
895 measurements (same as standalone ADL cases) suggesting there was no significant under/oversampling or evaporation of test
particles at the pickup tube inside the IPV (Fig. 4). The particle transmission efficiencies shown in this section are equal to
$E_S \cdot E_L \cdot E_{PCI} \cdot E_T$ $(\simeq E_L \cdot E_{PCI})$.

PCI-C-PM$_1$ configurations for higher altitude sampling with larger $d_{CO,down}$ and hence lower $P_{PCI}$ were tested in this
study (Fig. 15a). Lower $P_{PCI}$ sharply reduced larger particle transmission, with $d_{va,50,high}$ being ~950, 650, 600, and 250 nm for
220, 300, 350 and 400 μm $d_{CO,down}$, respectively. The detailed characterization of PCI-C-PM$_1$ configuration ($d_{CO,up}/d_{CO,down}$ =
350/220 μm) and comparison with other aerosol instruments during the NASA ATom mission were described in Guo et al.
(2021). During the ATom mission, $d_{va,50,high}$ of the inlet was ~750 nm for ATom 1–3 and ~950 nm for ATom-4 (Guo et al.,
2021), respectively. For PCI-C-PM$_{2.5}$, the trend was similar but the upper-end particle transmissions, $d_{va,50,high}$ were ~1500 nm,
1000 nm, 750 nm, and 400 nm (for the same set of $P_{PCI}$), were higher than PCI-C-PM$_1$ due to the improved large particle
transmission of the PM$_{2.5}$ lens (Fig. 15b). This suggests that, for PCI-C-PM$_1$, the $d_{va,50,high}$ is mainly limited by $E_L$ of the PM$_1$
lens especially when $P_{PCI}$ is higher ($d_{CO,down}$ is smaller). In the PCI-C-PM$_{2.5}$ setup, $d_{va,50,high}$ is mainly limited by the particle
transmission of PCI-C since all the measured $d_{va,50,high}$ with PCI were worse than that of the standalone PM$_{2.5}$ lens. Thus, for
further improvement of the overall inlet TE with a PM$_{2.5}$ lens, the PCI transmission needs to be significantly improved. Fig.
15c shows the transmission efficiency of PCI-D (designed as a part of this work) with a PM$_{2.5}$ lens. Due to unknown reasons,
PCI-D required EV$_{up}$ downstream of CO$_{up}$ to enhance the transmission efficiency of large particles. Unlike PCI-C, using larger
CO$_{down}$ only marginally reduced $d_{va,50,high}$ (~1370, 1300, 1100 nm for 125, 96, 72 mbar $P_{PCI}$). For the inlet configurations with
the PM$_1$ lens and PM$_{2.5}$ lens coupled with PCI-C, D, the $d_{va,50,low}$ was nearly identical to that of a standalone lens (Fig. 15a–c)
indicating the diffusion losses of small particles in PCI-C, D (30–100 nm $d_{va}$) were negligible.

The optimal operation altitude ranges and their $d_{va,50}$ ranges of the multiple PCI-ADL combinations in this work and
previous studies are compared in Fig. 15d (see Table S11.2 for PCI setups and reported TEs). Here, optimal operating altitude
is defined as the altitude where the ambient pressure is sufficiently high to ensure that the pressure upstream of CO$_{up}$ is higher
than $P_{PCI}$, so that the PCI can maintain its nominal $P_{PCI}$. The upper limit of the optimal operating altitude is determined by the
set PCI pressure and the lower limit is determined by the pumping capacity of the PCI pump. Note that the input pressure at
the inlet can differ slightly from ambient pressure. For example, when the sampling inlet is located over the wing, the sampling
line pressure can be 50-100 mbar lower than ambient. An additional 50–100 mbar pressure drop occurs inside the sampling
line. However, the pressure upstream of the PCI (sampling line pressure) tends to be higher than the ambient pressure during
flights due to the ram pressure on the sampling inlet into the aircraft. These additional effects do impact on the actual
operational altitude (Fig. S17.2).

The "PCI II" with PM$_1$ lens from Bahreini (2008) was one of the first PCI designs deployed for AMS aircraft operation
with optimal operation up to ~6 km ($P_{PCI}$ = 467 mbar). In that study, while $d_{va,50,low}$ and $d_{va,50,high}$ were not measured, TE was
near unity in the range of 150–650 nm $d_{va}$. Thus its $d_{va,50}$ range is wider than what is shown in Fig. 15d. Schmale et al. (2010)
deployed the same PCI design with different $d_{CO,up}/d_{CO,down}$ setups (380/250 μm and 400/160 μm with $P_{PCI}$ = 110 and 387 mbar,



respectively). While the 110 mbar $P_{PCI}$ setup ("PCI 3") had a good $d_{va50}$ range (80–1000 nm) up to ~15 km altitude, $P_{PCI}$ was not kept constant below 4 km altitude due to pumping limitations. The 387 mbar $P_{PCI}$ setup ("PCI POLARCAT") had a
narrower $d_{va,50}$ range (130–450 nm) but could maintain the nominal $P_{PCI}$ from sea level up to ~7 km altitude.

The PCI-C-PM$_1$ inlet was deployed during the NASA ATom and FIREX-AQ campaigns maintaining nominal $P_{PCI}$ up to a pressure altitude of about 9 km. The PCI-C-PM$_1$ configuration increased both the $d_{va,50}$ range, and operational altitude range compared to the previous inlets. While not ideal, this configuration guaranteed that even at maximum NASA DC-8 altitude (~12.5 km), the pressure in the PM$_1$ aerodynamic lens stayed above 1.3 mbar, and hence the aerodynamic focusing
into the vaporizer was not substantially impacted. Rather, overall TE appeared to be slightly improved in such conditions potentially due to sub-isokinetic sampling at the secondary diffuser during ATom (Guo et al., 2021). Under these conditions, the flow rate in the ADL decreases resulting in a lower sampling rate and hence slightly lower sensitivity of the AMS. Thus, an inlet that can maintain a constant $P_{PCI}$ (and hence a constant $P_{Lens}$) at the maximum altitude at a given aircraft campaign is desired.

Unlike the PCI designs discussed above, the constant pressure inlet (CPI) controls pressure upstream of ADL with a single pinched orifice by adjusting the diameter of the orifice actively during a flight (Molleker et al., 2020). Laboratory experiments showed a decreasing trend for $d_{va,50,high}$ at lower ambient pressures (*i.e.* larger orifice diameter) and TE vs. $d_{va}$ has a more gentle slope than for the PCIs (Fig. 15c). $d_{va,50,low}$ for CPI was not measured. Overall, $d_{va,50,high}$ range of CPI is larger than other inlets, especially at lower altitudes, but comparable to PCI-D-PM$_{2.5}$ at higher altitudes.

The PCI-D-PM$_{2.5}$ has significantly expanded $d_{va,50}$ and operation altitude range compared to the previous PCIs with the PM$_1$ lens. With the PCI-D with 450/350 μm ($d_{CO,up}/d_{CO,down}$) configuration used initially during the TI$^3$GER campaign, a triple-stage diaphragm pump (Vacuubrand Model MD1) was not sufficient to keep $P_{PCI}$ down to 72 mbar below 1.7 km altitude (Fig. S17.1). For operations including sea level altitude, a 450/300 μm ($d_{CO,up}/d_{CO,down}$) configuration (as used during the second part of TI$^3$GER) or a 450/350 μm ($d_{CO,up}/d_{CO,down}$) configuration with a more powerful (and heavier) PCI pump can be used.
Shattering of monodisperse NH$_4$NO$_3$ particles — used as a standard calibrant for AMS sensitivity — have been observed when operating earlier PCI designs (Guo et al., 2021) as well as single critical orifice setups. These shattering issues were not observed with either PCI-C or PCI-D.

Fig. 15e compares the detectable size distribution of particles in the lower stratosphere for a "typical" stratospheric distribution (an average of ATom data in the LS) using different inlet setups characterized in this study. By applying the
measured TE, we estimate that the PCI-C-PM$_1$, PCI-C-PM$_{2.5}$, and PCI-D-PM$_{2.5}$ configurations (at $d_{CO,down}$ = 300 μm and $P_{PCI}$ = 122.6 mbar) would deliver to the AMS vaporizer approximately 65%, 78%, and 89% of the ambient particles mass, respectively. Therefore, PCI-D-PM$_{2.5}$ was used for the aerosol measurement up to upper troposphere and lower stratosphere (UTLS) during TI$^3$GER. The residence time inside the PCI was calculated and measured using NH$_4$NO$_3$ particles (Sect. S12). Given the smaller overall size and lower operating pressure, the residence time inside the PCI was shortened by approximately
half (0.2–0.5 s) compared to the PCI-C-PM$_1$ deployed during ATom campaigns (0.5–1 s) (Fig. S12.1), reducing potential diffusional and evaporative losses.





**Figure 15: TE measurements of (a) PM$_1$ lens coupled with PCI-C (PCI-C-PM$_1$) (b) PM$_{2.5}$ lens coupled with PCI-C (PCI-C-PM$_{2.5}$) and (c) PM$_{2.5}$ lens coupled with PCI-D (PCI-D-PM$_{2.5}$) at various $P_{PCI}$ (various $d_{CO,down}$). For 220, 300, 350, and 400 μm $d_{CO,down}$, $P_{PCI}$**

**were set to 250, 125, 96, and 72 mbar, respectively (see Table S11.2 for $d_{CO,up}$ sizes), in order to maintain the nominal $P_{Lens}$ (2 mbar for PM$_1$ lens and 5.1 mbar for PM$_{2.5}$ lens). Solid lines in (a–c) are eye guidelines for the measured data using trapezoidal shapes in log($d_{va}$) space. (d) $d_{va,50,low}$ to $d_{va,50,high}$ range, and optimal operational altitude range of inlet systems characterized in this study and in previous studies. The $d_{va}$ range where TE ~ 1 is shown for "PCI II" with PM$_1$ lens since $d_{va,50}$ was not measured. The $d_{va,50,low}$ of CPI with PM$_{2.5}$ lens was not measured. The $d_{va,50,high}$ of PCI-C-PM1 from Guo et al. (2021) is 750 nm for the ATom 1–3 campaigns (as**

**shown in the figure) and 950 nm for the ATom 4 campaign. (e) TEs of tested aircraft inlet systems and their measurable particle distribution in the lowermost stratosphere vs. geometric diameter ($d_{geo}$). The size distribution shown is an average of all the stratospheric data (using the criteria discussed in Koenig et al, (2020)) reported by the NOAA Aerosol Microphysical Properties**





**instrument during NASA ATom 1–4 campaigns (Brock et al., 2019). Aerosol density was assumed to be 1.75 g cm$^{-3}$ since the majority of the stratospheric particle composition is sulfuric acid with some aged organics.**

## 3.5. Aircraft inlet performance during the TI$^3$GER campaign

### 3.5.1. Particle beam diagnostics in the field

Four sets of 2D-SR-BWP measurements were performed to monitor and optimize the beam alignment of the PCI-D-PM$_{2.5}$ inlet during the TI$^3$GER campaign, with the polydisperse aerosol generation setup (Fig. 1). The 2D-SR-BWP beam profile measurements were performed on four different non-flight days (Fig. S16.1a). Overall, the beam widths of the PCI-D-PM$_{2.5}$ system as a function of $d_{va}$ were similar during the TI$^3$GER campaign (Fig. S16.1c). The beam relative positions were nearly identical for BP1–BP3, while BP4 showed noticeable changes in the BWP X axis (Fig. S16.1d). This indicates that the beam pointing by the PCI-D-PM$_{2.5}$ as a function of $d_{va}$ remained nearly identical except for BP4. The exception is probably due to tilting of the inlet plumbing due to additional strain. The relative beam position of the PM$_{2.5}$ showed similar position variability in the X axis while the position was relatively constant in the Y axis.

After each flight (except for FF01 and RF05), 2D-BWP measurements with monodisperse NH$_4$NO$_3$ particles (350 nm $d_m$) were performed to track the relative change in lens alignment and fill the gap between the 2D-SR-BWP measurements. In Fig. 16a, the four profiles from 2D-SR-BWP were adjusted so that the beam of the same $d_{va}$ as the monodisperse beam (from the flights that were conducted after each 2D-SR-BWP) is matched with the position of the monodisperse beam. These adjusted BP1–4 were assigned to be the BPs of RF01, RF02-RF03&FF01, RF04, and RF05-RF08 & FF02, respectively. There are clear shifts in beam profile positions on the vaporizer. These variabilities can be attributed to various factors such as the installation of AMS on aircraft, vibration during flights/landing, and inlet remounting after service including replacement or cleaning of critical orifices (Fig. S16.1a). The inlet was serviced during the TI$^3$GER campaign since this topic was under direct investigation. Such shifts would not be expected for other research missions.

The particle beam model (Sect. 2.3.3 and Sect. 3.3) was used to estimate the particle transmission efficiency that accounts for the losses due to the particles failing to hit the vaporizer (Fig. 16b). For all the 2D-SR-BWP measurements, the decreasing trend of TE at $d_{va} < 80$ nm is captured by the model (consistent with PCI-D-PM$_{2.5}$ TE). For RF01, compromised TE ($d_{va} < 250$) nm is due to the beam profile of the size range being too close to the vaporizer edge while the compromised TE in the 400–700 nm $d_{va}$ range is mainly due to a broad beam width in X axis (Fig. S16.2a). The in-field TE measurement after RF01 at 688 nm $d_{va}$ was ~ 0.75, consistent to modeled TE (Fig. 16b). Overall, the particle losses due to failing to hit the vaporizer are negligible for the other flights during the TI$^3$GER campaign. In general, the aircraft inlet for AMS during the TI$^3$GER campaign worked well as expected from the laboratory experiment, despite several inlet adjustments. The in-field 2D-SR-BWP measurements and modeling suggest that the particles downstream of the PCI-D-PM$_{2.5}$ inlet were mostly well captured by the vaporizer.



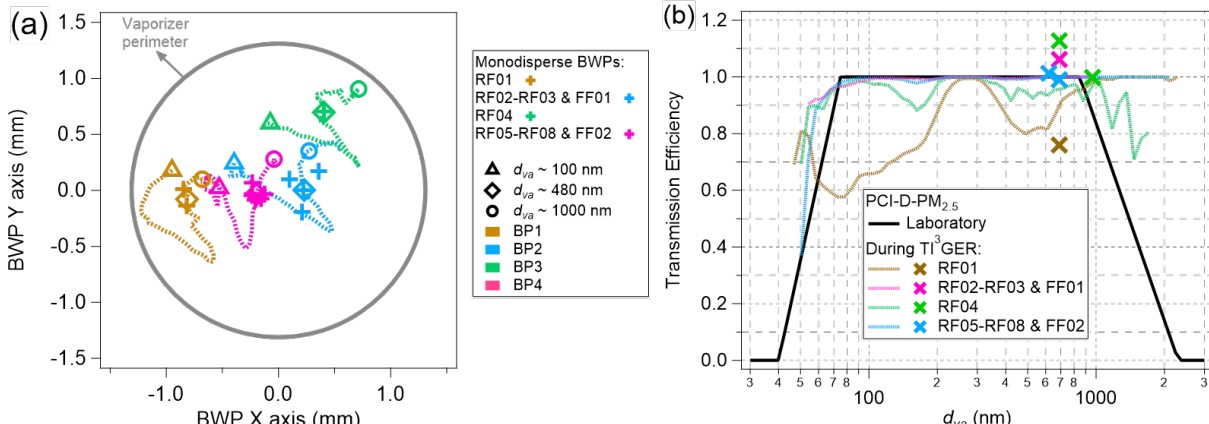


**Figure 16: (a) Compilation of beam center positions as a function of $d_{va}$ measured from 2D-SR-BWP analysis during TI³GER campaign (dashed lines). The grey solid line is the vaporizer perimeter at the BWP plane. The beam positions of monodisperse NH₄NO₃ particles are shown in cross markers after flights. The dotted lines represent the four beam profiles (BP1–4) (after being corrected to match with the monodisperse beam positions measured from research flights after each 2D-SR-BWP, see Fig. S16.1 for**
**details). (b) TEs modeled from the four 2D-SR-BWP (dashed lines) as well as the measured TE from the laboratory (solid line) of PCI-D-PM₂.₅. X markers are the measured TEs during the TI³GER with monodisperse NH₄NO₃ aerosols. Note that the model does not account for the particle losses inside or upstream of ADL.**

### 3.5.2. AMS vs. UHSAS aerosol volume comparison

The time series of calculated aerosol volume concentration measured by AMS ($V_{chem}$) and by UHSAS ($V_{phys}$) volume during

the TI³GER campaign are shown in Fig. 17a. The aerosol volume from AMS and UHSAS for this subset of the data agreed

very well (slope = 1.00, $R^2$ = 0.96, Fig 17b, see the comparison in log scale in Fig. S19.2). For this comparison the UHSAS

volume was corrected for estimated RI that was estimated based on the AMS composition (the originally reported data was

for an RI of 1.595). In this comparison, we only used data above 3 km altitude (to minimize the mass fraction of non-refractory

sea salt aerosols) and the data in the absence of clouds indicated by the NCAR cloud droplet probe instrument (Droplet

Measurement Technologies Inc.). Within the marine boundary layer, sea salt aerosols can be externally mixed and may not be

fully dried before the detection by UHSAS or AMS making the quantification and comparison more complex. One particular

period below 3 km altitude when the G-V sampled a transported pollution plume (yellow shaded area in Fig. 17a bottom panel)

over the Pacific Ocean was included in the intercomparison since the period was dominated by non-seasalt aerosols.

Given the focus of the development in this paper, TI³GER LS data was analyzed separately. In Fig. 17c, the fraction

of particles sampled by AMS was estimated in the LS where $H_2O/O_3 < 20$ (Koenig et al., 2020). For the estimation, the

measured distribution was corrected by the estimated particle losses in the UHSAS sampling line. The particle transmission of

the AMS aircraft inlet was applied to the corrected distribution. With the correction, the AMS is expected to sample 99%

(sampling line loss not considered) and 89% (sampling line loss considered) of the ambient accumulation-mode mass. If the

PCI-C-PM₁ inlet ($P_{PCI}$ = 122.6 mbar) were used during the TI³GER campaign, 63% of the mass would have been sampled.

Thus, the aircraft inlet for AMS during the TI³GER campaign is suitable for the quantitative sampling of accumulation mode

aerosols in the UTLS as well as other environments where a larger accumulation mode is expected (*i.e.*, urban haze events).





At the same time, the inlet's capability of sampling Aitken mode aerosols (*e.g.*, particle formation and growth events) is not significantly compromised vs. the aircraft inlet during ATom campaigns (Fig. S11.3).




**Figure 17: (a) Time series of aerosol volume concentrations from the AMS ($V_{chem}$) and the UHSAS ($V_{phys}$), altitude, and the estimated real part refractive index of the ambient aerosol in 1 minute resolution. (b) Scatter plot of AMS vs. UHSAS aerosol volume. The comparisons in log scale and altitude dependence are shown in Fig. S19.2. (c) Ambient aerosol size distribution measured by the UHSAS and the AMS detectable size distribution. The AMS sampling line loss calculation is shown in Fig. S18.1. Note that during**

**the campaign, there were several minor issues with the PCI-D-PM$_{2.5}$ inlet such as critical orifice clogging during RF01 and failure to maintain the set $P_{PCI}$ value during RF03 and RF04 in the marine boundary layer due to PCI pump limitation in 450/350 μm ($d_{CO,up}/d_{CO,down}$) configuration (Fig. S17.1). These issues do not affect the intercomparison between AMS and UHSAS in this section since UHSAS was not working properly during RF01 and the MBL during RF03 and RF04 were not included in the comparison. The PCI-D successfully maintained the set $P_{PCI}$ at higher altitudes including the highest altitude 13.8 km (Fig. S17.1).**




### 3.5.3. Particle transmission upstream of PCI during aircraft measurements

The airspeed into the tip of the secondary diffuser is controlled by the main flow rate which is actively controlled based on altitude, with slower aspiration speeds used at higher altitudes to keep losses at the initial 90 degree bend at a minimum while roughly matching the airspeed inside the HIMIL, assuming a slowdown within the HIMIL of about 6. The airspeed inside the HIMIL can be estimated based on the measurements at the USAFA wind tunnel facility performed after the TI$^3$GER campaign described in the SI (Sect. S20). Based on those measurements, the HIMIL slowed down the flow by a factor of ~7.4, as a function of air speed (Fig. S20.6). The airspeed at the tip of the secondary diffuser was 90–120% the extrapolated airspeed inside the HIMIL. This means that aerosol over/undersampling at the secondary diffuser is unlikely. This also suggests that, in future campaigns, the flow through the secondary diffuser could be reduced to maximize $E_T$ while maintaining the near isokinetic sampling inside the HIMIL. To our knowledge, this provides the first experimental air slowdown measurements for the HIMIL inlet. However, the experimental conditions covered in the tunnel experiments only match that of flights on jet aircraft up to 5 km altitude or so (Fig. S20.3). This means that the extrapolation of the curve toward the faster external air speed (higher altitude) is more uncertain.

The particle transmission through the sampling line ($E_T$) downstream of the HIMIL and upstream of the PCI-D that accounts for the diffusion and impaction losses are estimated based on the tubing dimensions and flow rate (Sect. S18) using the model described in Guo et al. (2021) and Bourgeois et al. (2022). The main loss of particles occurs in the 4.57 mm ID stainless tubing with a cumulative bending angle of 335 deg before the pickup tube (Fig. 2, Sect. 2.6 and Fig. S20.1), including the 90-degree bend at the secondary diffuser. At higher altitudes, the reduced pressure inside the sampling line enhances the impaction losses due to higher Stokes numbers. $d_{va,50,high}$ of the sampling line ranged 1.7–2.3 μm $d_{va}$, depending on the altitude (up to 14 km altitude). $d_{va,50,high}$ of the whole aircraft inlet (including PCI-D-PM$_{2.5}$) ranged within 1.3–1.5 μm $d_{va}$.

### 4. Summary and conclusions

In this work, we present a) the development of diagnostic tools (lens scan and 2D-SR-BWP) for measuring the beam width and pointing of collimated aerosol beams, b) the observation of position-dependent decomposition using the lens scan technique, c) measurements of $E_L$ and diagnostics of beam focusing and pointing with different Aerodyne ADLs (PM$_1$ lens, PM$_{2.5}$ lens, and HPL) using 2D-SR-BWP and beam modeling, d) the characterization of the newly developed PCI (CU PCI-D) and the aircraft inlet for quantitative aerosol sampling at high altitudes and its particle transmission efficiency tested in the laboratory, and e) an analysis of the performance of the improved aircraft inlet and diagnostics tools during the TI$^3$GER campaign. A characterization of the air flow inside the HIMIL inlet by wind tunnel experiments is also presented in the supplemental information. We also present advances in test aerosol preparation. Modification and characterization of an evaporation-condensation system allowed the measurement of transmission efficiency of small particles (30–200 nm $d_{va}$) without multiply charged particles. The modified 3D-printed nebulizer produced polydisperse NH$_4$NO$_3$ particles with a wide size range (40 nm to 4 μm $d_{va}$) for 2D-SR-BWP analysis.





A lens scanning stage was developed for optimization of ADL lens alignment. The lens scan technique finds the center and perimeter of the vaporizer quickly and accurately with monodisperse aerosols. Lens scans also proved useful to measure narrow particle widths ($< 0.25$ mm) by analyzing the signal decay at the vaporizer edge. Lens scanning also provides a useful tool for the investigation of particle-vaporizer interactions. Both for $NH_4NO_3$ and oleic acid aerosol, a higher degree of thermal decomposition has been observed when the particle beam impacts near the center of the SV relative to the off-center position. These higher degrees of thermal decomposition are probably caused by the additional collisions of vaporized gases with the vaporizer surface due to the conical shape of SV. This effect may have contributed to the previously observed instrumental variability of $f_{44}$ (Crenn et al., 2015; Fröhlich et al., 2015). Frequent calibrations are essential for monitoring and correcting for these variabilities (*i.e.*, the ratio of ratio method for organic nitrate apportionment; Day et al. 2022). Due to its broader particle beam width, deployment of a $PM_{2.5}$ lens can reduce the position-dependent decomposition effect, but at the cost of a more complex alignment procedure due to the variability in beam focusing.

The 2D size-resolved BWP (2D-SR-BWP) method was developed using polydisperse aerosols ($NH_4NO_3$ in this study) in PToF mode along two orthogonal axes. This technique provides 2-dimensional information of particle beam position and width vs. $d_{va}$. Beam widths measured by 2D-SR-BWP were consistent with BWP measurements with monodisperse particles including small sizes that are hard to access with monodisperse measurements. This technique revealed that the particle beam focusing (width) and pointing (center position) can vary as a function of $d_{va}$ for all the ADLs tested. For all lenses, the beam width was wider for smaller particles (*i.e.*, below 100 nm $d_{va}$) leading to lower $E_L$. The $PM_1$ lens and HPL showed a monotonic decrease in beam width for increasing $d_{va}$, as narrow as 0.05 mm or below. The $PM_{2.5}$ lens exhibited beam widening around 400–600 nm $d_{va}$. The particle beam position for the $PM_{2.5}$ lens and HPL varied noticeably depending on $d_{va}$, while the variability in beam position was minimal for the $PM_1$ lens. The particle beam model based on 2D-SR-BWP measurements can estimate the particle losses by failing to impact the vaporizer due to beam broadening and/or off-centered beam pointing, as a function of $d_{va}$. The particle transmission estimated from this method was consistent with measured transmission in the laboratory. The beam model also reasonably reproduced the particle deposition images.

$E_L$ of the $PM_1$ lens, $PM_{2.5}$ lens, and HPL were measured and compared with literature values. The $d_{va,50}$ range measured from the $PM_1$ lens was 43–940 nm. The small particle transmission of the $PM_1$ lens was very similar to previously reported values (Zhang et al., 2004a; Knote et al., 2011) and higher than the measurement by Liu et al. (2007). The measured $E_L$ of the $PM_{2.5}$ lens was good for both small and large particle sampling ($d_{va,50}$ range from 55–2700 nm). The measured $d_{va,50,low}$ in this study was 55 nm, 95 nm smaller than the $d_{va,50,low}$ from Xu et al. (2017). Given the $d_{va,50}$ range measured in this work, the $PM_{2.5}$ lens can be an excellent option for $PM_{2.5}$ sampling without significantly compromising the sampling of Aitken mode aerosols. However, due to the irregularity of beam focusing and pointing by the $PM_{2.5}$ lens, careful optimization of lens alignment and frequent beam monitoring is recommended to fully take advantage of their wide particle transmission range, especially so when combined with a CV. When diagnostic tools (*e.g.*, BWP) are not available, regular checks on the $E_L$ of the $PM_{2.5}$ lens can be beneficial. The $d_{va,50}$ range of HPL in this work was 96 nm to $> 2$ μm, consistent with Williams et al. (2013). The HPL proved to be outstanding in sampling super micron particles and submicron particles with some compromise in Aitken mode



particle sampling. At a lower lens pressure than typical operation (thus lower lens flow and lower sensitivity), the PM$_{2.5}$ lens and HPL showed lower $d_{va,50,low}$. Operation at low lens pressure may be useful when nanoparticle sampling (*i.e.*, new particle formation and growth events) is of interest.

A new design for the pressure-controlled inlet (PCI-D) was developed for quantitative sampling at high altitudes (up to 15–17 km) with minimal particle losses in the PCI. TE of PCI-C and PCI-D coupled with the PM$_1$ lens and PM$_{2.5}$ lens were measured in the laboratory. Since the $d_{va,50,high}$ of PCI-C-PM$_1$ is limited by $d_{va,50,high}$ of the PM$_1$ lens (rather than the PCI), deploying the PM$_{2.5}$ lens increases the $d_{va,50,high}$ of the inlet. As expected, operating at lower $P_{PCI}$ (for operations at higher altitudes while maintaining the nominal $P_{Lens}$) led to higher particle losses (lower $d_{va,50,high}$), while $d_{va,50,low}$ was not significantly affected. The addition of the expansion volume (EV-D$_{up}$) downstream of CO$_{up}$ and the modification of the expansion volume

downstream of CO$_{down}$ (EV-D$_{down}$) significantly improved TE while operating at low $P_{PCI}$ (below 250 mbar). Despite the improved TE of the PCI, $d_{va,50,high}$ of PCI-D-PM$_{2.5}$ is still limited by the particle transmission of PCI-D. The PCI-D-PM$_{2.5}$ lens ($d_{va,50}$ range 60–1400 nm, $P_{PCI}$ = 122.6 mbar) is suitable for quantitative measurements of accumulation mode aerosols in the UTLS. Compared to the previous inlet configuration used for CU-HR-AMS (PCI-C-PM$_1$), the increase in operation altitude and $d_{va,50,high}$ were 5 km and 500–700 nm, respectively. The operation at 96 and 72 mbar enables the operation of PCI-D at

even higher altitudes (up to 16 km and 18 km, respectively) with similar transmission as the 122.6 mbar setup if the PCI pump is capable of maintaining the constant $P_{PCI}$ at lower altitudes. The residence time in the PCI-D is 0.2–0.5 sec which is ~50% of the PCI-C. The PCI-D-PM$_{2.5}$ is expected to measure ~90% of the accumulation mode aerosol mass in the lowermost stratosphere during the ATom campaign without compromising the flow through the ADL. The aircraft inlet system coupled with a PM$_{2.5}$ lens is useful for better quantifying aircraft-based aerosol sampling in other environments with larger

accumulation aerosol in general (*e.g.*, urban haze event).

    The aircraft inlet was tested during the TI$^3$GER campaign, up to ~13.8 km altitude over the northern Pacific Ocean. The in-field 2D-SR-BWP measurements and beam modeling provided the estimation of the particle losses due to the irregular beam focusing and pointing depending on $d_{va}$, which was not significant except for RF01. The PCI-D maintained the constant pressure in the intermediate-pressure volume region (IPV) at the highest altitude. A good volume closure between AMS and

UHSAS in the free troposphere and the lower stratosphere during the TI$^3$GER campaign was observed. The particle transmission of the aircraft inlet was ultimately limited by the particle losses in the sampling line upstream of PCI-D-PM$_{2.5}$. During the TI$^3$GER campaign, the AMS aircraft inlet sampled 89% of accumulation mode aerosol mass in the UTLS, mostly limited by the sampling line, and not the PCI or ADL. The post-campaign measurements of air speed inside the HIMIL reveal that the air slow-down ratio is ~7.4. For future deployments, the flow through the secondary diffuser can be reduced while

maintaining near isokinetic sampling inside the HIMIL, ultimately reducing the particle losses in the sampling line upstream of PCI and improving the overall transmission of larger particles.



**Data availability.** TI³GER AMS and UHSAS data can be found at https://doi.org/10.26023/QFEJ-E81T-DC0W and https://doi.org/10.26023/CNDV-BZJ3-880X, respectively. (Last access: 6 December 2022) Data for all the figures in the paper
(including SI) can be downloaded from https://cires1.colorado.edu/jimenez/group_pubs.html.

**Author contributions.**

DK, PCJ, HG, JLJ designed/conducted experiments and analyzed data on ADLs and PCIs. DY, SD ran/analyzed CFD and contributed in designing PCI-D. DK, PCJ, RV, JLJ designed and performed HIMIL wind tunnel testing. DK conducted particle
beam simulations. DK, PCJ, HG, DAD collected AMS data during the TI³GER campaign. DK, PCJ performed the TI³GER data reduction. MR collected UHSAS data. RV designed and led the TI³GER campaign. LW, JJ, DW provided aerodynamic lenses and experimental data. LW and PC acquired particle disposition images. All authors contributed to writing the manuscript.

**Competing interests.** LW, PC, JJ and DW are employees of Aerodyne Research, manufacturer of the AMS and the ADLs used in this work.

**Acknowledgments.** We thank Charles Brock for the loan of the LAS instrument and acknowledge him, Agnieszka Kupc and Christina Williamson for providing the size distribution measurements for ATom. We thank Sarah Woods and Michael Reeves
for collecting CDP data and Michael Reeves for collecting UHSAS data. TI³GER was supported by the National Center for Atmospheric Research, which is a major facility sponsored by the NSF under Cooperative Agreement no. 1852977. The data were collected using NSF's Lower Atmosphere Observing Facilities, which are managed and operated by NCAR's Earth Observing Laboratory. The GV aircraft was operated by the National Center for Atmospheric Research (NCAR) Earth Observing Laboratory's (EOL) Research Aviation Facility (RAF). Wind tunnel testing was conducted at the US Air Force
Academy Aeronautics Research Center under Commercial Test Agreement 21-161-AFA-01. We thank Melissa Morris and Masayuki Takeuchi for manuscript review and useful discussions. We thank the machine shop at the University of Colorado Boulder Chemistry Department for manufacturing the custom designed PCIs.

**Financial support.** This research has been partially supported by NSF grant AGS-2027252, and NASA grants
80NSSC18K0630, 80NSSC19K0124, 80NSSC21K1342, 80NSSC21K1451, and 80NSSC23K0828. DK has also been partially supported by the CIRES Graduate Student Research Award and the AGU Jerome M. Paros Scholarship in Geophysical Instrumentation.

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
