# Peer review of "Development and Characterization of an Aircraft Inlet System for Broader Quantitative Particle Sampling at Higher Altitudes: Aerodynamic Lenses, Beam and Vaporizer Diagnostics, and Pressure-Controlled Inlets"

_Aerosol Research, 2025_

## Author Response (AR1)

**Responses to Reviewers' Comments on the paper "Development, Characterization and Rapid Diagnostics of an Aircraft Aerosol Mass Spectrometer Inlet System"**

We thank the two anonymous reviewers for their comments. Below, we address the reviewer comments in black text, and our responses follow in blue text. Changes to the text are shown in **blue**, **bold text**.

**Responses to Reviewer 1:**

**R1.1.** Aerosol Mass Spectrometer Inlet System" by Kim et al. is a very detailed study on the focussing properties and the transmission efficiency of aerodynamic lenses used for the Aerodyne AMS. Additionally, it describes the aircraft inlet system required to maintain a constant pressure in the aerodynamic lens (pressure-controlled inlet, PCI) and the transmission efficiency of the combination of CPI and aerodynamic lens. An automated procedure to scan the lens position and thereby the particle beam over the vaporizer was designed and is presented.

We thank the reviewer for their time, positive review, and the detailed suggestions for improvements, which we address below.

**R1.2.** The title therefore is not fully adequate, to my opinion it should also reflect the findings on the lens focussing properties, the particle beam width, and the transmission properties of the lens types. Also the consequences for the analysis (especially ion ratios for organic nitrates) are important.

Thus, the paper is more on "Characterization of focussing and transmission efficiencies of aerodynamic lenses and aircraft inlet systems used for an Aerosol Mass Spectrometer, and development of a rapid diagnostics tool"

We agree that the original title may have been too narrow, and that it needs to better represent the diagnostics tools and aerodynamic lens characterization more clearly. Hence, we changed the title to:

**"Development and Characterization of an Aircraft Inlet System for Broader Quantitative Particle Sampling at Higher Altitudes: Aerodynamic Lenses, Beam and Vaporizer Diagnostics, and Pressure-Controlled Inlets"**

We also revise the abstract (line 24-26) to read:

"These techniques allow for fast automated aerosol beam width and position measurements and ensure the aerodynamic lens is properly aligned and characterized for accurate quantification, in particular for small sizes that are hard to access with monodisperse measurements. The automated lens alignment tool also allows investigating position-dependent thermal decomposition on the vaporizer surface."

**R1.3.** General remark: Given the importance of the correct positioning and alignment of the lens, would it be useful to design a better fixing mechanism such that the lens is adjusted once and then remains in its optimal position?

In principle, yes, although the lens stage lock is already strong. However, note that other parts can move as well (*e.g.*, the differential pumping cylinder around the vaporizer and the vaporizer itself. Also, it is possible that the pieces inside the lens might shift slightly, leading to the observed changes. As with the original Aerodyne design, the issue is more about resistance to prolonged vibrations (e.g. long road shipping on a truck), which are hard to address by something that is still supposed to be adjustable.

Also, please note that the changes in the beam profiles in Fig. 16 are mostly due to unlocking the inlet for servicing and locking it again, rather than the inlet moving itself. The intention in showing this figure is to showcase the ability to do this in the field frequently - we are not suggesting in any way that this is needed for a normal campaign, and indeed, this was not an issue in the recent ASIA-AQ campaign.

However, it is true that the locking mechanism could use some improvement since the locking process can, at times, result in some translation of the lens.

R1.4. line 46-48: Sentence not correct, maybe skip "that" in line 46?

Revised. Yes, that was a typo.

**R1.5.** line 54: Maybe add reference to DeCarlo et al 2004/5 for d\_va? That reference also explains how to convert d\_mob into d\_va.

We already show the relationship between  $d_{va}$  and  $d_{mob}$  in Eq. 5 and cite Decarlo et al. (2004) there.

**R1.6.** line 75 "the SV"?

Revised.

**R1.7.** line 170: 0.05 M ?

Revised.

**R1.8.** line 175: Fig S7.1 shows the size distributions. From the text I had expected to see the nebulizer itself.

To address the reviewer's comment, we added the cross-section of the 3D-printed nebulizer in Fig. S7.1 as below:

Figure S7.1. (a) Cross-section of the 3D printed nebulizer in Rösch and Cziczo (2020). (b) A modified 3D printed nebulizer used in this work. The diameter of the main chamber is 1.111 cm in the original design and 1.715 cm in our modified version, which uses different fittings to interface with the rest of our setup and the critical orifice mount discussed - but not shown - in Rösch & Cziczo. (c) Mass size distribution of  $NH_4NO_3$  particles generated by the customized 3D printed nebulizer (b) (blue) and the TSI atomizer (model 3076, red). Most of the measured distributions were within ~ 100% transmission efficiencies of the inlet used for each nebulizer. The nebulizers were operated at ~ 35 psi nozzle pressure.

**R1.9.** Line 191: In recent years, it was recommended (at AMS user meetings) to use the mass based IE calibration. Did you do that and did you compare it to the ET mode IE calibration results?

Aerodyne's recommendation for general users, particularly since the introduction of the ACSM (which lacks ET capability), is and has been to use mass-based IE. This is due to the higher likelihood of mistakes by inexperienced users when using the ET or BFSP method. The Jimenez Group group has been using BFSP/ET for two decades for AMS calibration and have found it to be faster and more reproducible than mass based calibrations, as documented in Guo et al, 2021. BFSP/ET is completely independent of an external measurement and hence still works reliably in the event that the transmission of the chosen calibration particle size is <100%.

In this paper, the fact that we achieve ~ 100% transmission efficiency with ammonium nitrate particles (*e.g.*, Fig. 14 & 15) shows that ET based IE and mass based IE calibrations were consistent, since the transmission efficiency was calculated based on AMS mass and CPC mass of monodisperse particles (Eq. 11). Also, the fact that the particle volume between AMS and UHSAS agreed during the TI3GER campaign (Fig. 17) indicates that the ET based IE was consistent with the UHSAS aerosol quantification.

We are currently finalizing a manuscript (Nault et al., in prep) that describes both calibration methods and discusses the relative benefits in detail. It is certainly true that while there are many studies detailing AMS mass calibrations, ET/BFSP calibrations are not that well documented in the literature.

To address the reviewer's comment, we changed the text (line 191) to read:

"AMS sensitivity to nitrate was calibrated with monodisperse  $NH_4NO_3$  single particles in event trigger (ET) mode (Decarlo, 2009; Kimmel, 2016). Compared to mass-based calibration, ET calibrations are independent of any inlet losses (plumbing or otherwise). Note that mass-based IE calibration is more straightforward to carry out and generally recommended. The fact that ~ 100% of transmission efficiency is measured with  $NH_4NO_3$  particles (Table S9.4) indicates that IE from ET calibration is consistent with that from mass-based calibration."

**R1.10.** Line 219: Should read Table S9.3. What do you mean by "air signal delay"? The "PToF"-signal of the m/z 28?

Yes, we meant the time delay of the air signal between chopper open and measurement by AMS. To clarify, we revised the text to read:

**" $v_g$ (by measuring the time of flight of the air signal at m/z 28)"**

**R1.11.** Line 239, Equ. 6: The unit conversion factor is only needed if V\_chem is given in  $\mu$ g sm-3, right? The equation would be correct w/o the conversion factor if you replace all densities by "rho\_species" (as you did for "rho\_OA"). The values for rho can be explained in the following sentence, as you did for seasalt. Where does the value of 1.52 g cm-3 for Cl come from?

That's correct, the conversion factor is needed when aerosol species are in  $\mu$ g sm-3 units, density is in g cm-3 units, and aerosol volume is in  $\mu$ m3 sm-3 units. We prefer to keep the units of density to be g cm-3 since that is most widely used and intuitive, and hence keep the conversion factor.

The chloride density was adopted from Salcedo et al. (2006). See our response to R1.12 for the revised text.

**R1.12.** Line 243-245: You did not take into account rBC, because no measurement was available. But was a measurement for seasalt available?

Yes, fine-mode mode seasalt aerosols are quantified by AMS by using a calibrated response factor for the NaCl+ ion (Ovadnevaite et al., 2012) which is specific to seasalt. However, seasalt aerosol quantification is not as robust as other non-refractory aerosols due to the strong dependence on the particle transmission/response at the upper end of the AMS transmission curve and the fact that seasalt is typically externally mixed, which introduces additional uncertainty in the bulk volume calculation. Hence, for the comparison of total aerosol volume vs. UHSAS, we focus on periods above 3 km altitude to minimize the influence of seasalt aerosols, except for the period marked in Fig. 17a.

To address the reviewer's comments (R1.11 & R1.12), we revised the text (line 241) to read:

"Aerosol chemical components (µg sm-3) were measured by AMS. Seasalt density (1.45 g cm-3) was taken from Guo et al. (2021), assuming partially deliquesced particles (Brock et al., 2019).

Seasalt mass concentration was quantified following Ovadnevaite et al. (2012) using a custom calibration factor (1/110 vs. 1/55 in Ovadnevaite). However, the uncertainty in the fractional volume of seasalt due to mixing state (external vs. internal) and the strong sensitivity to the shape of the transmission curve are significant. For that reason, when comparing  $V_{chem}$  vs.  $V_{phys}$ , we focus on the altitude above 3 km where the seasalt influence is minimal. Density of non-refractory chloride was adopted from Salcedo et al. (2006), based on NH4Cl literature values."

R1.13. Line 357: better: 800 hPa (SI unit)

Revised.

R1.14. Line 370 ff/Fig.5/Table 1: what is the inner diameter of the EV-D\_up and EV-D\_down?

It's 16 mm for both. We revised the caption (line 407) of Fig. 5 to read:

"The  $P_{PCI}$  for CU PCI-D shown in the figure is when  $d_{CO,down} = 300 \ \mu\text{m}$ . The inner diameter of EV-Dup and EV-Ddown is 16 mm."

**R1.15.** Line 398: is "informed" the right word here? Does it mean "based on"? Or "The cone angle and dimensions... were chosen based on ..."?

Our coauthors, Da Yang and Suresh Dhaniyala, conducted numerical fluid dynamic modeling to simulate the aerosol flow field within the PCI in order to help find a better design with minimum particle losses. While the final design of EV-D\_down is indeed inspired by those results, the overall final design was pretty empirical and hence it seems appropriate to say that it has been informed by these model results.

R1.16. Line 429: HIMIL has already been introduced earlier.

Accepted. Now the reference is moved to when HIMIL is introduced for the first time and here the acronym is used.

**R1.17.** Line 446: -> "are reported"

Revised.

**R1.18.** Line 627: I thought m/z 57 is for HOA; not m/z 55?

Both m/z 55 and m/z 57 tend to be high in HOA factor (Jeon et al., 2023). However, we agree that mentioning COA instead of HOA is more appropriate here since  $f_{55}$  is more specific to COA, and the test aerosol compound used here (oleic acid) is mainly present in COA. At the same time, we will note that this exploration was mostly to get a sense how aliphatic ions are affected by positioning, and not about a specific primary type of OA. We revised the text (line 627) to read:

"On the other hand, m/z 55 fraction ( $f_{55}$ , mostly C4H7+), which is often used as an indicator of cooking OA, was not noticeably affected by the location of particle impact (Fig. S5.6e)."

**R1.19.** line 646: May it also be that no particles larger than 1.5  $\mu$ m d\_va were produced by the particle generator?

Fig. S7.1 shows that the nebulizer is capable of generating much bigger particles (measured in PToF mode with HPL), which is why we could use 2D-SR-BWP for the HPL lens. Also, size distribution measured from OPC supports this (data not shown). Thus, low signals above 1.5 um  $d_{va}$  for the inlet suggest that these particles are lost inside PCI.

We revised the text (line 646) to read:

"Above ~1.5 µm  $d_{va}$ , signal is limited by inlet transmission, with substantially decreasing signal to noise (note that the nebulizer aerosol output spans beyond 1.5 µm  $d_{va}$  as shown in S7.1c)."

R1.20. line 732: What is "ruby orifice"?

This has been described in a recent paper, which was published after our paper was submitted (Nault et al., 2025). According to that paper, "A new exit nozzle for the  $PM_{2.5}$  aerodynamic lens, manufactured from ruby rather than stainless steel, has been evaluated and provides more uniform and tighter focusing of the particle beam." We modify the text (line 732) in our paper to read:

"These features were also observed from a different Aerodyne's  $PM_{2.5}$  lens with improved nozzle design with ruby orifice (data not shown). More details about the  $PM_{2.5}$  lens with ruby exit nozzle can be found in Nault et al. (2025)."

**R1.21.** Fig 12b: Above 150 nm, model and measurement agree perfectly? That's too good to be believed.

The model constructs an aerosol beam based on the measured beam width and position from 2D-SR-BWP measurements (Sect. 2.3.3). Then, the transmission efficiency is the fraction of particles (in the model) that is captured by the vaporizer. We did not need to introduce any additional correction factors to match the modeled transmission and the measurement. However, as noted in the paper (line 791-795), the model tends to overestimate the transmission efficiency when the particle beam is located near the edge of the vaporizer.

**R1.22.** lines 835-843: Knote et al. (2011) do not mention explicitly a d\_50, the use a linear slope for the transmission between 0 and 100% vs log d\_va from 40 to 100 nm (and then the same for 550 - 2000). Please explain this, otherwise it's hard for the reader to find it in the Knote paper. Furthermore, in line 842 I would not say "recommended", I would say "used by".

Although  $d_{va,50}$  is not specifically mentioned in Knote et al. (2011), it can be calculated from  $d_{va,0}$  and  $d_{va,100}$ . Transmissions at  $d_{va,0}$  and  $d_{va,100}$  are linear in  $\log(d_{va})$  space. Therefore,  $d_{va,50}$  is the geometric mean of  $d_{va,0}$  and  $d_{va,100}$ . We revised the text to read:

"The measured  $d_{va,50,low}$  for the PM1 lens in this work ( $d_{va,50,low} \sim 47$  nm) was lower than the reported value by Knote et al. (2011) ( $d_{va,50,low} \sim 63$  nm, as calculated from the geometric mean of  $d_{va,0,low}$  and  $d_{va,100,low}$  in the paper) which is an averaged value from multiple studies ..."

**R1.23.** Fig 14, line 884: Molleker et al. (2020) used the ERICA mass spectrometer with an Aerodyne PM2.5 lens, not the ALABAMA which has the custom designed lens as described in Clemen et al., 2020. Please check and correct.

Yes, Molleker et al. (2020) used ERICA, not ALABAMA. Thank you for the correction. We revised the text (line 884) to read:

"Molleker et al. (2020) used the ERICA (ERC Instrument for Chemical composition of Aerosols; ERC – European Research Council) (Hünig et al., 2022) and CPI ..."

**R1.24.** line 861: dva,50,high ~ 120 nm, -> should read dva,50,low

Revised. Thank you for catching this typo.

**R1.25.** line 909-910: "Due to unknown reasons, PCI-D required EVup downstream of COup to enhance the transmission efficiency of large particles." I don't understand the meaning of this sentence. I understood that was the reason for adding the expansion volume EV.

The PCIs before PCI-D had only one EV downstream of  $CO_{down}$ . When PCI-D is operated at lower  $P_{PCI}$  pressures (in order to operate at higher altitude), it showed significant particle losses when only one EV (EVdown) was used. When another EV (EVup) was added upstream of  $CO_{up}$ , the particle transmission efficiency was improved very noticeably. However, we do not understand how or why EVup helped improve transmission efficiency. CFD simulations did not help clarify this mechanism. That is what we meant by the sentence. To make this point clearer, we revised the sentence to read:

"As discussed in Sect. 2.5, the addition of  $EV_{up}$  downstream of  $CO_{up}$  in PCI-D enhanced the transmission efficiency of large particles significantly, although the reason for the improvement is not clear."

R1.26. Fig 15 c), d): legend: please correct Mollecker -> Molleker

Revised. Thank you again for catching the typo.

**Responses to Reviewer 2:**

**R2.1.** This paper describes a new pressure-controlled inlet, designed for use with the Aerodyne AMS instrument, that allows both for operation across a range of tropospheric to lower stratospheric pressures and across a wide range of particle sizes as well as a computer-controlled alignments system. With numerous airborne AMS instruments, the community needs a highly capable inlet of this sort.

We thank the reviewer for acknowledging that the community needs more advanced aircraft inlets such as the one presented in this work.

**R2.2.** Unfortunately, this paper is not publishable in its current form. The paper is far too long. The authors' stated goal in the title and abstract is to characterize and describe a new inlet system and include performance from a set of flights; reading the paper, it is evident this could be accomplished by a paper cut by a factor of 5 or more in length. The main paper is 44 pages before the references with 17 figures. The supplement has a further 20 figures, many highly repetitive, and is another 58 pages of length. I will state this for clarity - there are currently 102 pages of manuscript with 37 figures to describe only a new inlet design.

As stated in the title and abstract, our goal of this paper is to develop and characterize the aircraft inlet system, as well as to develop and characterize the diagnostic tools that are useful to diagnose the inlet, ideally under field conditions. This is not "only a new inlet design". We regret that the scope of this paper was perhaps not clearly communicated by the original title. Since reviewer 1 raised a similar point (see our response to reviewer comment R1.2) we agree that the title needs to be revised to better represent the major topics covered in this paper. Hence, as mentioned in the response to R1.2, we changed the title to prevent such potential confusion by other readers to:

"Development and Characterization of an Aircraft Inlet System for Broader Quantitative Particle Sampling at Higher Altitudes: Aerodynamic Lenses, Beam and Vaporizer Diagnostics, and Pressure-Controlled Inlets"

We also revise the abstract (line 24-26) to read:

"These techniques allow for fast automated aerosol beam width and position measurements and ensure the aerodynamic lens is properly aligned and characterized for accurate quantification, in particular for small sizes that are hard to access with monodisperse measurements. The automated lens alignment tool also allows investigating position-dependent thermal decomposition on the vaporizer surface."

Additionally, to reduce the length of the main text, we have moved Section 3.3 titled "Validation of particle beam model with TE and particle deposition" and the two associated figures (Fig. 12 and 13 in the original preprint) to the supporting information and merged with Section S8. The section numbers and figure numbers that appear in the main text and the supporting information were revised accordingly.

**See our response to R2.3 for further discussion.**

**R2.3.** I recommend rejection. The authors should cut this paper to be both impactful and readable; this journal is not meant to be a vast repository of all information they collected during their development process. If the focus, as stated, is the design and development of the inlet then please maintain that but eliminate all the extraneous information and repeated figures. I suspect their objectives could be met with a paper length of something like 15 pages and 5 or 6 figures. At that time a normal review cycle could take place.

Reviewer 2 recommended rejection solely based on the length of the paper, without providing specific criticisms of the content. As discussed in the response to R2.2, we believe that this may arise from a misunderstanding about the scope of this paper. Again, this manuscript is not about "only a new inlet design". Our manuscript aims to provide a full description of the new aircraft inlet system by offering characterizations of its components (HIMIL, PCI, aerodynamic lens, vaporizer, in the laboratory and in the field), as well as aerosol beam diagnostics tools that we developed that have proved crucial to our team, and that can be very useful for other applications in the AMS community.

We will note, for example, that currently there are only 2 published (and inconsistent) studies detailing the  $PM_{2.5}$  lens transmission on the ground, at least partly because of the complexity of the calibration procedure. This adds large uncertainties in urban studies that could be reduced by using the 2D-BWP method. Therefore, we respectfully disagree and believe that all the information presented in the paper is separately valuable.

We understand that our manuscript is very technical, complex, and long for general readers. However, this work is targeted at readers who are interested in precise aerosol quantification and airborne measurements using AMS (or other mass spectrometers that utilize aerodynamic lenses), and we believe the information will prove useful to them. In any case, scientists do not generally read papers linearly, but they typically zero in on the sections of interest.

Finally, we would like to point out that papers with comparable preprint lengths to our preprint (45 pages after moving a section as discussed in the response to R2.2, and excluding references) have been published in Aerosol Research and Atmospheric Measurement Techniques, such as Aliaga et al. (https://doi.org/10.5194/ar-2024-15, 45 pages, excluding references), Mak et al. (https://doi.org/10.5194/egusphere-2024-1232, 46 pages, excluding references), and Meyer et al. (https://doi.org/10.5194/egusphere-2024-2021, 47 pages, excluding references). Reading the reviews of these papers, their length was not raised as a critical issue for any of them. Although the length of our preprint is indeed on the longer side, manuscripts of this length are not unusual. Roughly, each issue of AMT (which publishes mostly instrument-related papers and thus is more comparable with this manuscript) has one manuscript of length similar to ours (Fig. R1).

Figure R1. Histograms of the number of pages (excluding references) in the preprints published in (a) Aerosol Research since the start of journal publication and up to Volume 3 Issue 1 and (b) Atmospheric Measurement Techniques Volume 18 Issue 1-5.

**References:**

[revised manuscript text omitted]